

# A high-fidelity multiresolution DEM model for Earth systems

Duan Xinqiao[1], Li Lin[1, 2, 3], Zhu Haihong[1, 3], Ying Shen[1, 3]

[1]Geographical Information Science Faculty, SRES School, Wuhan University, Wuhan, 430079, China
[2]Geospatial information science Collaborative Innovation Centre of Wuhan University, Wuhan, 430079, China
[3]The Key Laboratory for Geographical Information System, Ministry of Education, Wuhan, 430079, China

*Correspondence to*: Li Lin (lilin@whu.edu.cn)

**Abstract.** The topographic impacts on modifying Earth systems variability have been well recognised. As numerical simulations evolved to incorporate broader scales and finer processes, accurately embedding the underlying topography to simulate land – atmosphere – ocean interactions, or performing commensurate scale transformation to topography while considering high-fidelity retention have proven to be quite difficult. Numerical schemes from Earth systems either use empirical parameterization as sub-grid scale and downscaling skills to express topographic endogenous processes, or rely on insecure point interpolation to induce topographic forcing, which create bias and input uncertainties. DEM generalisation provides systematic topographic transformation by considering loyal fidelity, but existing heuristic approaches are not performed optimally because of point clustering, or are difficult to incorporate into numerical systems because of sliver triangles. This article proposes a novel high-fidelity multiresolution DEM model with high-quality grids to meet the challenges of scale transformation. The generalized DEM model is initially approximated as a triangulated irregular network (TIN) via selected terrain feature points, control points, and possible embedded terrain features. The TIN surface is then optimized through an energy-minimized centroidal Voronoi tessellation (CVT). By devising a robust discrete curvature as a density function and exact geometry clipping as an energy reference, the developed curvature CVT (cCVT) converges, the generalized model evolves to a further approximation to the original DEM surface, and the points and their dual cells become equalized with the curvature distribution, exhibiting a quasi-uniform high-quality grid. The cCVT model is then evaluated on real LiDAR-derived DEM datasets compared to the classical heuristic method. The experimental results show that the cCVT multiresolution model outperforms classical heuristic DEM generalisations in terms of both surface approximation precision and surface morphology retention.

## 1 Introduction

### 1.1 Topography in Earth systems

Topography is one of the main factors controlling processes operating at or near the surface layer of the planet (Florinsky and Pankratov, 2015; Wilson and Gallant, 2000). With the success of Earth and environment systems in conforming theories, explaining observations with these scale diversity processes, there exists persistent demands for extending their utility into new and expanding scopes (Ringler et al., 2008; Tarolli, 2014; Wilson, 2012). The pushing demand may originate from the





requirements for simulating processes and scales beyond the current numerical scheme, and may also emerge as response for migration from coarse gross estimation to fine regional predictions of environmental systems, as exemplified by lapse-rate controlled functional plant distributions (Ke et al., 2012), orographic forcing imposed on oceanic (Nunalee et al., 2015) and atmospheric dynamics (Brioude et al., 2012; Hughes et al., 2015), fine-grained topographic relief dominated extreme

hydrological processes as flood inundations (Bilskie et al., 2015; Hunter et al., 2007), and many other geomorphological (Wilson, 2012), soil (Florinsky and Pankratov, 2015), and ecological (Leempoel et al., 2015) examples from different components of Earth systems. However, as numerical systems evolved to incorporate broader scales and finer processes to produce more fidelity predictions (Ringler et al., 2011; Weller et al., 2016; Wilson, 2012; Zarzycki et al., 2014), how to accurately embed the underlying topography with increasingly improved resolution (with the popularity of airborne or

terrestrial LiDAR technology) to simulate land-atmosphere, land-ocean, or land-hydrology interactions, or how to perform commensurate scale transformation to the topography itself taking care of the coupling endogenous features have proven to be a quite difficult task (Bilskie et al., 2015; Chen et al., 2015; Tarolli, 2014).

Climate or weather simulation models usually adopt sub-grid scheme to exert topographical heterogeneity and rely on downscaling of the finer-resolution climate observations to surface variables (Fiddes and Gruber, 2014; Kumar et al., 2012;

Wilby and Wigley, 1997). Coupled or assimilated climate observations construct a reasonable base for dynamic or statistical downscaling, which increases the model resolvability to broader scales. However, reliant atmosphere or climate observations themselves are always of confined resolution, whilst sub-grid surfaces are designed to accommodate empirical parameterization rather than full feature capturing, which implies bias of endogenous lateral-variability representation and mixed-up grid cell of uncertainties (Jiménez and Dudhia, 2013; Nunalee et al., 2015). The static boundary conditions, i.e.,

topographic relief, are also commonly embedded through point interpolation in atmosphere-land-ocean interaction simulations, and mesh refinements are used to handle dynamic boundary conditions and minimize topographic source errors (Guba et al., 2014; Kesserwani and Liang, 2012; Nikolos and Delis, 2009; Weller et al., 2016). However, mesh from interpolated points does not necessarily comply with the terrain relief, and underlying elevation errors are frequently reported as one input uncertainty (Bilskie and Hagen, 2013; Hunter et al., 2007; Nunalee et al., 2015; Wilson and Gallant,

2000). While there are many situations exist where dynamic conditions are stressed for their stronger impacts on modifying prediction results than static topography conditions (Budd et al., 2015), even in the topography-driven flood processes where refinements are dynamically imposed on wet/dry fronts (Cea and Bladé, 2015; Nikolos and Delis, 2009), but the underlying topographic layer is still prominently important for its increasingly improved fidelity to the Earth's surface and vast prospective in widening application scenarios (Bates, 2012; Tarolli, 2014), and a sophisticated topography transformation

treatment would be beneficial by minimizing discrepancies arisen from physical inconsistencies (Chen et al., 2015; Glover, 1999; Ringler et al., 2011).



## 1.2 Multiresolution DEM model

Systematic scale transformation of topographic data considering loyal fidelity has long been studied under terrain generalisation, where precise surface approximation and terrain structural feature retention have both been pursued (Ai and Li, 2010; Chen et al., 2015; Guilbert et al., 2014; Jenny et al., 2011; Weibel, 1992; Zhou and Chen, 2011). Triangulated

irregular networks (TIN) are generally chosen as a substitute for regularly spaced grids (RSG), and critical points or salient points with terrain significance are selected for the network feature points. Triangular networks are used for their adaptiveness to a locally enhanced multiresolution scheme. Critical points or salient points are selected because they can effectively improve the approximation precision (Heckbert and Garland, 1997; Zakšek and Podobnikar, 2005; Zhou and Chen, 2011).

As surface approximation precision and terrain feature retention are competitive for the redistribution of feature points, DEM (digital elevation model) generalisation is differentiated from terrain generalisation for its emphasis on surface approximation as a whole with the aim of providing best surface interpolation, but shares a mutual goal of detail reduction for clearance of confusions and distractions with terrain generalisation (Guilbert et al., 2014). Terrain generalisation emphasises geomorphology or landform depiction where (map) cognitive efforts are drawn to produce progressive data

reduction, with the effects that the main relief features strongly stressed while non-structural details are massively suppressed (Ai and Li, 2010; Guilbert et al., 2014; Jenny et al., 2011). Since the static topographic conditions are commonly composed directly by DEM datasets to diverse simulation interests, maintaining precise surface approximation for rigid boundary conditions is more important than 'sparse' geomorphological feature abstraction. While DEM datasets are usually used interchangeably with topography or terrain in Earth systems, we will use DEMs and topography indiscriminately

hereafter.

Existing DEM generalisation can be roughly catalogued into two groups, namely, heuristic refinements and smooth fitting, according to differences in the surface approximation strategy. The first class of approaches is due to the computational bottleneck consideration, that determination which combination of vertices to a TIN mesh approximates the original DEM surface best needs exponential time (Chen and Li, 2012; Heckbert and Garland, 1997). It thus forces existing methods to

25 employ some heuristic strategy, which adopts greedy insertion refinements (or deletion) on feature points, to find a sub-optimal approximation solution that is computationally practical. In each insertion (or deletion), rearranging the entire existing mesh to obtain a better approximation is also computationally inhibitive and thus not adopted (Chen et al., 2015; Heckbert and Garland, 1997; Lee, 1991), and this may result in the clustering of feature points. Among those existing heuristic approaches, trenching the pre-extracted terrain features (drainage streamlines mainly) into the TIN mesh seems

quite appealing (compound method) (Chen and Zhou, 2012; Zhou and Chen, 2011), but the quality of the generalized TIN mesh cannot be guaranteed, and the existence of thin sliver triangles makes it difficult to be incorporated with numerical stability (Kim et al., 2014; Weller et al., 2009). The second class of approach is due to the consideration of TIN mesh from feature points do not necessarily warrants best approximation to the original dense DEM, for feature points are commonly





selected through some local metric. Many research thus considered surface approximation globally instead of elaborated feature point selection, such as bi-linear, bi-quadric, multi-quadric, Kriging, or general radial base function-based fittings (Aguilar et al., 2005; Chen et al., 2015; Schneider, 2005; Shi et al., 2005). The proposed multi-quadrics (Chen et al., 2012; Chen et al., 2015), for example, approximates the original DEM surface well with a high-order smooth surface and the

5 smooth surface provides a kind of weeding mechanism to cure the feature point clustering problem. But the high-order radial base function is computational expensive when a broad scenario is involved (Chen et al., 2015; Mitášová and Hofierka, 1993). In brief, existing DEM transformations are neither best performed with loyalty to the original terrain surface, nor easily incorporated into numerical schemes.

The purpose of this article is to devise a multiresolution DEM model that considers optimized surface approximation and

10 guaranteed high-quality surface grids. The high-quality grid is demanded by surface interpolation precision and easy cooperation with digital systems. Multiresolution is an effective paradigm to model scale diversity; different ways to multiresolution models have been proposed and tested by many studies in recent years, in terms of both variable-resolution modelling and grid refinements, such as block-structured grids, cubed spheres, centroidal Voronoi tessellations (CVTs) (Du et al., 1999; Du et al., 2010), nesting and r-refinement skills (Guba et al., 2014; Ringler et al., 2011; Weller et al., 2016).

Among those promising plans, we are especially fascinated by the CVT's capability for redistributing sample points, and its nature way for smoothing transient areas which are unattainable for blocked structures or situ h-refinement skills (Du et al., 2010; Ringler et al., 2011). Based on DEM generalisation and shape evolution, we develop CVT to an optimized surface approximation model to realize multiresolution terrain scheme.

### 1.3 Outline of the proposed method

The generalized terrain surface is initially approximated by a triangular grid that is constructed from selected feature points and then optimized through centroidal Voronoi tessellation. The selected feature points include points of interest to embed field control points or ground checking points. The construction of an initial approximation surface also incorporates terrain features of interest. The CVT is driven by a robust discrete curvature as density function, based on the curvature's ability on shape characteristics capturing and shape evolution (Banchoff, 1967; Kennelly, 2008; Pan et al., 2012). This curvature CVT

(cCVT) at the first step makes feature points spatially equalized along the 2-dimensional surface through an approximated Voronoi tessellation, a dual operation. By devising an exact geometry clipping-based energy estimation for every computed Voronoi cell, the cCVT at the second step calculates better representative positions through the curvature integral over the area and set as new Voronoi sites. The exact clipping and energy referring avoids some key numerical issues accompanied with classical clustering approaches, and makes it a total different method to the CVTs (Du et al., 2010; Valette et al., 2008).

The two steps of cCVT stabilize feature points at the spot where they and their dual Voronoi cells best approximate the original surface, and distributed as curvature density function in an iterative and energy-minimized manner. Thus, this cCVT method exhibits distinctive features compared to existing approaches. This method avoids strong reliance on feature points from local metric computation because of the global surface approximation strategy, and converges to a further





approximation of the original high-resolution terrain surface because of the surface optimization process, thus outperforms most existing heuristic DEM generalisations. Additionally, this method promises a quasi-uniform high-quality grid still with control points or features of interest embedding because of the spatially equalized feature points along the 2-dimensional surface. These along together build a high-fidelity multiresolution terrain model to incorporate more scales. The high-fidelity
multiresolution terrain model builds a reliable boundary layer for environmental simulations, upon which surface variables can be estimated under a coupled system, or computational mesh can be constructed and refined to possible dynamic conditions, with commensurate scale settings.

### 1.4 Organization of the article

The rest of the article is organized as follows. In Section 2 the theory behind CVT energy minimization iteration for
generalized DEM surfaces is introduced, techniques for incorporating DEM generalisation principles and fast convergence are presented, and the differences between the cCVT method and classical CVT clustering approach are discussed. In Section 3, the cCVT model is tested with real LiDAR-derived terrain datasets to evaluate surface approximation precision and grid quality, the experimental datasets description and comparison method qualification are also presented. Section 4 discusses the cCVT's considerations, comparable results, underlying causes, and interpretations. Finally, Section 5 presents
short conclusion and outlooks briefly.

### 2. Curvature centroidal Voronoi tessellation on DEM surface

### 2.1 Definition

Centroidal Voronoi tessellation is a space tessellation for each Voronoi cell's geometrical centre (in the spatial domain) that coincides with its barycentre from the abstract property domain (Du et al., 1999). Here, the property domain is analogous to
the frequency domain. For the vertices set t $\{v_i\}_k^1$ in $\Omega \subset R^3$, the Voronoi tessellation graph is defined as:

$$V_i = \{p \in \Omega : |p - z_i| < |p - v_j|, j = 1..k, j \neq i\}, i = 1..k. , \tag{1}$$

That is, a Voronoi cell $V_i$ is the set of points whose distance to $v_i$ is less than that to any other vertices. $|\cdot|$ is the Euclidean norm. Every vertex and its corresponding dual cell commonly embed an intensity scalar $\rho$ from some abstract property domain, which is called a density function. The total potential energy of the Voronoi graph $V$ of a terrain surface can be
computed by summing up every cell $V_i$'s potential energy:

$$E = \sum_{i=1}^k \iint \rho \cdot |p - v_i|^2 \cdot d\sigma , \tag{2}$$

Energy minimizer:

$$\bar{x} = \frac{\iint x \cdot \rho \cdot d\sigma}{\iint \rho \cdot d\sigma}; \ \bar{y} = \frac{\iint y \cdot \rho \cdot d\sigma}{\iint \rho \cdot d\sigma}; \ \bar{z} = \frac{\iint z \cdot \rho \cdot d\sigma}{\iint \rho \cdot d\sigma} , \tag{3}$$





which minimizes the surface's total potential, for $\overline{v}_i = (\bar{x}, \bar{y}, \bar{z})$ satisfies:

$$\frac{\partial E}{\partial p} = 2(p - v_i) \cdot \iint \rho \cdot d\sigma = 0 \ , \tag{4}$$

In other words, when $\boldsymbol{v_i}$ coincides with the barycentre $\overline{\boldsymbol{v}}_i$, each cell's potential effect on the property domain (gravity) becomes equalized to a stable energy state.

## 2.2 Lloyd Iteration

The most classical energy minimization process of centroidal Voronoi tessellation is expressed by *Lloyd's Relaxation* (Lloyd, 1982). The main idea of this algorithm is to first tessellate the surface; density integration over the area is performed to find a 'gravity' barycentre for each tessellated cell, which is then used as the new site for the iteration. The pseudo code of this procedure is shown below:

*Algorithm1 Lloyd_relaxation*
*Inputs: vertices set $\boldsymbol{N} = \{\boldsymbol{v_i}\}_k^1$*

*while ( $\Delta \boldsymbol{E} > \boldsymbol{Threshold}$ )*
*{*
    *use the $\boldsymbol{k}$ vertices to tessellate the surface, obtain Voronoi cells $\{\boldsymbol{V_i}\}$;*
    *clear $\boldsymbol{N}$ ;*

  *for each $\boldsymbol{V_i}$ in $\{\boldsymbol{V_i}\}$*
   *{*
    *Compute barycentre of $\boldsymbol{V_i}$: $\bar{x} = \frac{\iint x \cdot \rho \cdot d\sigma}{\iint \rho \cdot d\sigma}$; $\bar{y} = \frac{\iint y \cdot \rho \cdot d\sigma}{\iint \rho \cdot d\sigma}$; $\bar{z} = \frac{\iint z \cdot \rho \cdot d\sigma}{\iint \rho \cdot d\sigma}$;*
    *push ($\overline{\boldsymbol{x}}$, $\overline{\boldsymbol{y}}$) to $N$;*
   *}*
  *compute $\boldsymbol{E}$ ;*
*}*

We follow Lloyd's elegant idea. The barycentre of a 2-dimentional Voronoi cell may fall outside this surface patch, so an additional calculation may be needed to amend this. Du et al. suggested to project the barycentre onto a nearest facet and the constrained projection point is used instead (Du et al., 2003). Others suggest quadric interpolations over all the facets of the cluster for further accurate site calculation (Valette et al., 2008).

## 2.3 Fast converge to DEM equilibriums

*Lloyd's Relaxation* requires Voronoi tessellation on a discrete 2-dimensional surface, but direct Voronoi tessellation on a piece-wise smooth surface requires complicated geodesic computation and may be challenged by complex numerical issues (Cabello et al., 2009; Kimmel and Sethian, 1998),Du et al. suggested that CVT could be realized through some clustering means (Cohen-Steiner et al., 2004; Du et al., 1999; Du et al., 2003), that is, using some kind of attached property as density function to cluster facets and then find the clustered cells' barycentres to create new clustering sites. Through this heuristic



iteration, the new sites along with the new tessellations compose better and better approximation to the original surface, with their spatial distribution conforms to the pre-defined density function.

The clustering approach avoids geodesic tessellation by direct facets combination, which is computationally light. The greatest expenditure then comes from global distance computation for identifying every cell to its cluster centre. However, k-means like clustering over discrete facet set suffers from some key issues concerning Voronoi cells such as zigzag boundaries – since no geodesic Voronoi tessellation really used, and invalid clusters with disconnected set of facets which may introduce heavy clean-up operations (Valette and Chassery, 2004; Valette et al., 2008).

Terrain surface critical points such as peaks, pits, and saddles are treated as gravity equilibria and key elements depicting the surface geometry in the large (Banchoff, 1967; Milnor, 1963); a further extended critical points on a second-order surface derivative (such as curvature) will describe a more detailed set of terrain surface parameters (Jenny et al., 2011; Kennelly, 2008). When constructing a generalized DEM surface, these feature points are commonly used as a base to effectively improve the surface approximation precision and embedded with additional field control points, ground check points, or pre-extracted terrain structures for further approximation (Guilbert et al., 2014; Zakšek and Podobnikar, 2005; Zhou and Chen, 2011). These ground control points, check points, or pre-extracted terrain structures of interest are also required in numerical simulation setups for cross-checking, validation and confirmation purposes. Hence, terrain feature points can be used as natural approximations of the final CVT energy equilibria.

Based on the above observations and requirements, this article proposes a sample point set scheme (including boundary control points, feature points, and pre-extracted structural points of interest) as initial Voronoi sites. For optimized spatial distribution of these sample points and optimized surface approximation precision, we calculate a robust discrete mean curvature as density function, which is based on the recognition of curvature's flexibility on capturing shape characteristics and capability conducting shape evolution (Banchoff, 1967; Kennelly, 2008; Pan et al., 2012). Curvature's flexible ability on depicting terrain morphology has been appreciated by many researches. For example, P. J. Kennelly pointed out that, compared to hydrologic model, curvature-generated drainage networks has more nature performance, has not limit to one single pixel thickness, has no requirement on flat filling (depressions are rather useful sometime such as flash flood modelling), and capable of delineating both convergent flow and divergent flow (Kennelly, 2008). The robust discrete curvature calculation is referred to *Meyer et al.* (2003). From the sample set an initial TIN surface is constructed, we compute its dual mesh and take the space tessellation of the dual vertices as approximate Voronoi cells. But different to clustering approach, we use each approximate Voronoi cell to clip on the original DEM surface, called referring patch. By this exactly clipped referring patch we compute accurate energy estimation for new approximated sites. The global clustering computation is thus localized (a kd-tree is utilized for the trick) and accurate referring energy computation makes iteration converge fast. And more important, we successively approximate Voronoi tessellations but avoid problematic clustering. The localization makes geometrical operation costs minimized, the efficiency of the cCVT approximation as a whole is comparable to that of the elegant clustering approach. We go no further for the complexity analysis but however provide an



implementation of the classical clustering with the same settings as the cCVT in the attachment. The pseudo-code of this cCVT iteration is described as follows:

*Algorithm2 cCVT_iteration*

*Input: vertices $N = \{v_i\}_k^1$, scale transformation **Ratio**.*

*1) Construct the original DEM surface **oriPd** from vertices N, compute density function $\rho$ based on robust mean curvature estimation;*

*2) Extract and mark boundary points **B**, mark stationary control points, check points as **C**, extract and mark the feature points **F**;*

*3) Perform constrained Delaunay triangulation on point set **{B, C, F}**, with boundary **{B}** and structural terrain features **{C}** as constraints;*

*obtain an initial approximated **TIN** mesh;*

*4) While ($\Delta E > $ **Threshold**)*

*{      4.1) Compute **TIN**'s dual **TD**;*

       *4.2) For the **n** vertices $r_j$ in **TD**, extract its direct incident facets as $FS = \{T_i\}_n^1$;*

       *4.3) For each $T_j$ in **FS***

*{*

          *4.3.1) Compute its minimal bounding box **BBox**<sub>j</sub>, fast compute its intersection of **oriPd** using an kd-tree, obtain a narrowed reference geometry **narrPd**;*

          *4.3.2) Compute exact intersection of $T_j$ and **narrPd**, push the result into reference sets **REF={ref<sub>j</sub>}**;*

       *}*

*4.4) For each **ref**<sub>j</sub> in **REF***

       *{*

          *4.4.1) Compute approximated Voronoi barycentre: $\overline{x} = \frac{\sum \rho \cdot x \cdot area(ref_j)}{\sum \rho \cdot area(ref_j)}$; $\overline{y} = \frac{\sum \rho \cdot y \cdot area(ref_j)}{\sum \rho \cdot area(ref_j)}$; $\overline{z} = \frac{\sum \rho \cdot z \cdot area(ref_j)}{\sum \rho \cdot area(ref_j)}$;*

          *4.4.2) Use kt-tree for fast intersection computation of point $(\overline{x}, \overline{y}, \overline{z})$ and **oriPd**, with the result used as the projected nearest point; push it into the new candidate point set **F'**;*

*}*

       *4.5) Using **{B, C}** as constraints, Delaunay triangulate point set **{B, C, F'}** and obtain reconstructed **TIN'**;*

       *4.6) Compute **E** on **TIN'**;*

*}*

Here, we illustrate this algorithm by using a numerical model. The analytic equation of the selected mountain model is:

$$z = (4x^2 + y^2) \cdot e^{-x^2-y^2}, \tag{5}$$

It has two peaks, two saddles and a pit. We rasterize it with a 49×49=2401 regular grid (**Figure 1**, left). As for effectiveness, we set the generalized scale at 0.1 (**Ratio** = 0.1), that is, there are about 240 points left. The sample set includes 56 boundary points, 5 critical points, and an additional 169 random points for visual saturation purposes (**Figure 1**, left; the red points are

randomly generated points, the blue points are boundaries, and the green points are critical points). Relief feature points are



always abundant in a real terrain dataset, so additional random points are rarely needed. A robust mean curvature estimation is computed on the original high-resolution surface *oriPd* (**Figure 1**, right), by which we can clearly distinguish critical points as peaks, saddles, and pits. The initially approximated TIN surface from the sample set is shown in **Figure 2** (left), and its generated dual mesh is shown in **Figure 2** (right), which corresponds to step 3 of Algorithm2. Figure 3 shows the

dual cell of sample points, which is the key idea of the cCVT approximation. Figure 4 and Figure 5 shows the algorithm steps 4.3.1 and 4.3.2, respectively, where the exact clipping is completed on the original DEM surface. Figure 6 and Figure 7 show the final computation on the reference patch of the first sample point, which corresponds to the algorithm steps 4.4.1 and 4.4.2. **Figure 8** exhibits the result of the first iteration compared to that of the final iteration, with the initial sample points included (top). A comparison of the constructed approximate TIN meshes of the initial state and final state is

illustrated in the middle, while the curvature distribution that represents the terrain feature comparison is illustrated at the bottom.

The results show how the embedded stationary points (control points and boundary points), feature points, and random points are spatially equalized (**Figure 8**). Additionally, the cCVT generated a terrain-adaptive variable-resolution grid (middle right); the convergent TIN mesh exhibited nearly uniform high quality, and the convergence process generally

resembled *Lloyd's Relaxation* (Figure 9).

Notably, the cCVT iteration used a direct reference on the original DEM surface. The exact geometry clipping linearly interpolated the actual high-resolution surface, which guaranteed accurate energy estimation and avoided zigzagging Voronoi cells.

### 3 Multiresolution DEM Experiments

### 3.1 Experimental datasets

Two sites with significant geomorphological characteristics were selected. Experimental site 1 is Mount St. Helens, located in Skamania County, W.A., USA. This mountain is an active volcano whose last eruption occurred in May, 1980, and beneath deep magma chambers have recently been observed (Hand, 2015). This site was selected for its typical mountain morphology along cone ridges and evident fluvial features downhill, where heavy pyroclastic materials and deposits are

present. These two distinctively different terrain structures mingle together, posing challenges for DEM generalisation.

The   St.   Helens   dataset   was   selected   from   Puget   Sound   LiDAR (http://wagda.lib.washington.edu/data/type/elevation/lidar/st_helens/), this LiDAR dataset was collected in late 2002. The selected dataset is a 2924×3894 regular grid with a 3 m cell size and covers an area of approximately 102 km$^2$. The elevation ranges from 855.32 to 2539.34 m. The image and hillshade views of these data are illustrated in **Figure 10**.

Experimental site 2 is the Columbia Plateau, USA. This area has been labelled as American UTM zone 11, we hereafter call this   area   UTM11 (http://gis.ess.washington.edu/data/raster/tenmeter/), this LiDAR dataset was collected in 2009. The selected site is located on the border of Columbia County and Walla Walla County, WA. The south-eastern corner is located



in the Wenaha-Tucannon Wilderness, Umatilla National Forest. This area contains rugged basaltic ridges with steep canyon slopes at high elevations (average of 1700 m). The north-western area is located near Dayton City, which is a vast agricultural and ranching area with relatively smoother relief at low elevations (averagely 500 m). If the generalisation scheme emphasizes the prominent high elevation areas, the surface interpolation as a whole will be lost, which will smooth the features in the low elevation area.

The selected UTM11 dataset is a 3875 × 3758 grid with a 10 m cell size and covers an area over 1456 km$^2$. The elevation ranges from 3533 to19340 cm. The image and hillshade views of these data are shown in **Figure 11**.

### 3.2 Comparison method description

As aforementioned, DEM generalisation has long been studied in geoscience, with numerous methods proposed over time. One of the most classical approaches is the hierarchical insertion (or decimation) of feature points to construct a TIN grid under a destination scale. This type of heuristic feature point refinement (HFPR) performs very well in terms of surface approximation and terrain structure retention. For this reason, although HFPR methods generally cannot guarantee high-quality meshes, these methods are suitable for comparison purposes.

A typical HFPR starts with four corner points from a dense DEM image and constructs a Delaunay triangular mesh that contains two triangles. The rest of the points are weighted by their distance to the triangular surface or other error criteria and queued. The point with the largest priority in the queue is selected, and the mesh is modified by using incremental Delaunay triangulation. This process loops until some error threshold is satisfied (Heckbert and Garland, 1997). Michael Garland provided a classical HFPR implementation (http://mgarland.org/software.html), and many other variants are available in GIS, meshing, and visualisation tool suites.

### 3.3 Quantitative and qualitative comparisons

We performed the processes from Algorithm2 for the two experimental datasets, including triangulation and curvature estimation, boundary point extraction and marking, feature point extraction based on curvature significance and marking, shape optimization through cCVT, etc. The scale transformation ***Ratio*** was set to 1% (points left) to demonstrate the effectiveness.

The accuracy of the surface approximation determines the final surface interpolation precision and is thus a basic quality comparison index. An energy function is a suitable index for the CVT that represents the quality of the surface approximation, but this index is not usable for HFPR surfaces. Thus, we applied a statistical interpolation method to measure the surface approximation precision. From each facet triangle on the quasi-uniform mesh of the final cCVT-generated TIN grid, we introduced a vertical line that would intersect the original dense DEM surface and the HFPR generalized surface at the same time. We obtained an error estimation of the surface approximation from these intersection points. We computed the mean error, maximum error, and root mean squared error (RMSE) for this elevation interpolation (TIN interpolation); the





results are listed in Table 1. Furthermore, we computed the aspect ratios of the triangles for both generalized TIN meshes to measure the grid quality, which are also listed in Table 1.

A qualitative index is usually measured from terrain structure retention aspects. According to the generalized results from both experimental sites, both the cCVT and HFPR methods performed well in terms of the visual examination. However, upon closer inspection, the surface that was generated by cCVT has a smoother and more natural transition effect (Figure 12). Although HFPR accumulated more samples around sharp features (c.f. **Figure 13**), its surface looked clearer because relatively flat details were smoothed out. Under the same transformation conditions, HFPRs may exert a stronger generalisation effect than cCVTs. However, a stronger generalisation effect actually decreases the general approximation's precision, which may result in structural relief distortion or misconfiguration. **Figure 14** illustrates a closer examination of St. Helens. Some structural details on the original surface were recovered by the cCVTs but not by the HFPRs. This terrain structure loss occurred on both smooth areas and steep areas, as illustrated in **Figure 14**. **Figure 15** illustrates similar structural detail loss from HFPRs in the UTM11 dataset.

Terrain structural features could also be measured from DEM derivatives such as the slope, aspect, hydrological structural lines, etc. Here, we used contours to compare the generalisation accuracy using experimental site 2. Upon the same configurations (80 m elevation increments), we generated contours for the original dense TIN (rendered in red), the cCVT-generated TIN (rendered in blue), and the HFPR-generated TIN (rendered in black) and overlapped the three sets of contours for comparison (**Figure 16**). The illustrations demonstrate that, in most cases (**Figure 16** b, c), the contours from the cCVT-generalized surface more accurately conformed with those from the original dense surface, while the contours from the HFPR-generalized surface generally did not, except for some cases on steeper areas with sharp curvature variations (**Figure 16** d). This result can be explained by the HFPR's stronger accumulation of sample points on sharp features, which guaranteed an edging out, if we noticed that the inspection area **d** is much smaller than **b** or **c**.

## 4 Disscussion

Topography transformation that involves the simplification and generalisation of DEM surfaces has been a deeply studied topic in geoscience. Extracting terrain feature points and using these points to construct a generalized surface is one successful approach that is based on knowledge of the feature points' capability to capture terrain structures. However, TIN meshes that are constructed purely from feature points may not be the best approximation to original high-resolution (and high-accuracy) surfaces. Taking the mountain equation in **Figure 1** for example, it has at least two peaks, two saddles, and one pit close to zero level. Presume scale transformation requires that only two critical points are left; selecting both peak points is more reasonable than selecting the pit point, even if the pit point has a stronger quantitative index (curvature in this case) than those of the peak points. This observation implies that, if generalisation is considered from global interpolation statistics error aspect, a robust approach that incorporates surface approximation and terrain feature retention should be considered.



Among those classical DEM generalisation approaches, heuristic feature point refinement (here refinement is a method description opposed to decimation, rather than a mesh enhancement strategy) is an outstanding example. As illustrated by Table 1, **Figure 12**, and **Figure 16**, HFPR methods perform excellently in terms of surface approximation and morphology feature retention. On the treatment of feature points, these methods use a heuristics strategy by introducing incremental

Delaunay triangulation, which considers the point with the largest error from the constructed TIN. However, the impact of the feature point being-inserted on the feature points have-inserted is not considered because this requires the existing TIN mesh to be reconstructed, which would create a prohibitive computational burden. As a result, feature points may cluster around relief with sharp variations, as exemplified by **Figure 12**. Too many feature points accumulating near sharp features means that relatively scarce of feature points are present in flat areas, which would eventually lead to terrain structure

distortion or misconfiguration, as shown in **Figure 14**, **Figure 15**, and **Figure 16**. Sometimes, this type of structural loss is unbearable. For example, the terrain relief at high elevations under the studied scale (10 m cell spacing) in experimental site 2 indicated a fiercer landform than at lower elevations. The accumulation of too many sample points in high-elevation areas may result in the distortion or misconfiguration of the smooth anthropogenic terrain morphology in low-elevation areas.

cCVT starts by constructing a terrain-adaptive multiresolution grid. The iterations use a robust mean curvature as a density

function based on the curvature's capability to characterise shapes and conduct shape evolution. CVT is essentially an iterative two-step process (c.f. Algorithm1 in Section 2.2): Voronoi tessellation to equalize feature points in the spatial domain, and centroidal Voronoi computation to equalize feature points in the property domain. The process of spatially equalising feature points has been seldom considered by classical approaches, which may explain why cCVT generally prevails over HFPRs.

On the other hand, CVT is basically an approach within variational framework. The result from iteration relies on the boundary conditions and initial conditions. Hence, this article employed a feature point scheme (including boundary constraint points, field control points, and ground check points) as a relatively stationary initial condition to maintain algorithm stability. The requirement of embedding feature points of interest, along with consideration to avoid the problematic numerical issues concerning k-means clustering, prompt we to develop a non-clustering approach with an exact

energy referring method. Experiments on ten million DEM points demonstrated that the exact clipping approach performed comparably to the elegant clustering approach. Notably, the triangles from the cCVT-generated TIN mesh exhibited a maximum aspect ratio that was less than 5.0 (c.f. Table 1), which implies that the build-up terrain-adaptive mesh satisfied the numerical stability requirement from classical finite element or finite volume computations.

**5 Conclusions**

In this article, a high-fidelity multiresolution DEM model was proposed to meet the challenges of scale transformation. Multiresolution models are an essential tool to incorporate more scales such that a locally adaptive scheme can efficiently preserve scale-coupling features. A high-fidelity generalized DEM model can build a concrete topographic base from which



fine endogenous or exogenous processes can be derived under proper scale conditions. These two aspects were achieved by our devised curvature-based CVT. cCVT optimization increases the precision of surface approximations compared to most existing heuristic DEM generalisations, while the equalisation of feature points from both the spatial domain and curvature magnitude domain (i.e., frequency domain) facilitates multiresolution and high-fidelity approximations.

5 Evaluation of cCVT multiresolution DEM model on Earth and environmental systems in wide-ranged domains and scales is needed in further study. Considering the Earth system situation of global modeling tyranny (Ringler et al., 2011), this may imply a consideration of curvature of Earth itself into the cCVT model.

**Code Availability**

The main cCVT algorithm and the classical k-means clustering CVT implementation, which has the same building 10 environment as the cCVT, are provided. However, some source codes from the third parties were used in our research, and we do not have the rights to re-deploy these source codes. Please contact the corresponding author for the complete source code.

**Acknowledgement**

This study is funded by the Special Fund for Surveying, Mapping and Geo-information Scientific Research in the Public Interest (201412014), the National Natural Science Fund of China (41271453) and Scientific and Technological Leading Talent Fund of National Administration of Surveying, mapping and geo-information (2014).

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



**Table 1 Quantitative comparison of the mesh quality**

| Experimental LiDAR Dataset | Dense DEM Points | Interpolation Points | Method | Mean Error (m) | Max Error (m) | RMSE (m) | Max. Aspect Ratio |
|---|---|---|---|---|---|---|---|
| St. Helen 3 m | 11,386,056 | 230,909 | cCVT | 0.0353 | 23.05 | 1.61449 | 3.23 |
| | | | HFPR | 0.1107 | 191.31 | 2.3714 | 9255 |
| UTM11 10 m | 14,562,250 | 301,255 | cCVT | 0.5773 | 37.70 | 3.77313 | 4.09 |
| | | | HFPR | 0.8765 | 487.81 | 6.71214 | 8426 |





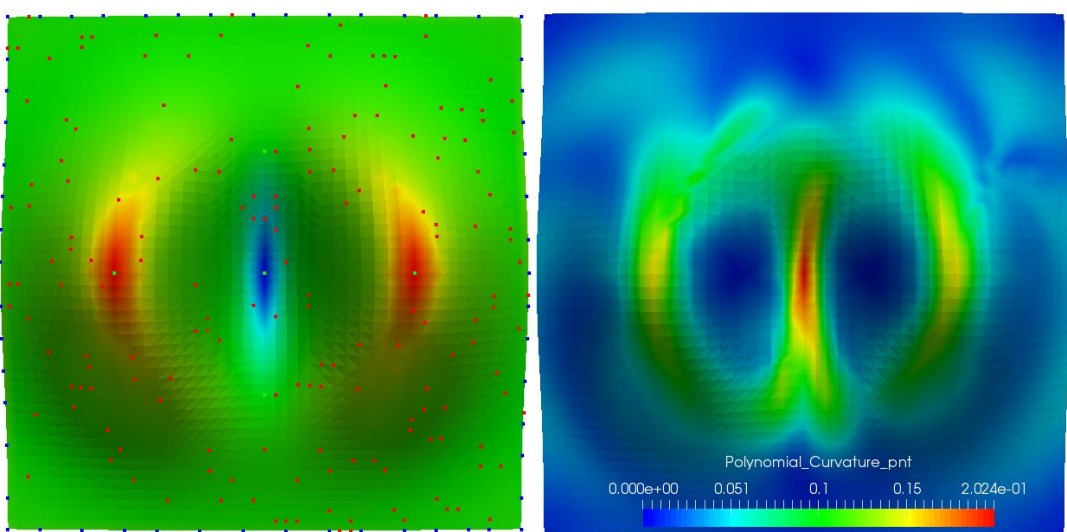

Figure 1  High-resolution grid of a mountain equation (5). Left: original mesh *oriPd* in 49×49 resolution, rendered in mean curvature. The sample points were also rendered on oirPd; the blue points are boundary points, the green points are critical points, and the red points are random points. Right: robust estimated mean curvature. The saddle terrain features, peaks, and pits can be more clearly distinguished compared to the left.

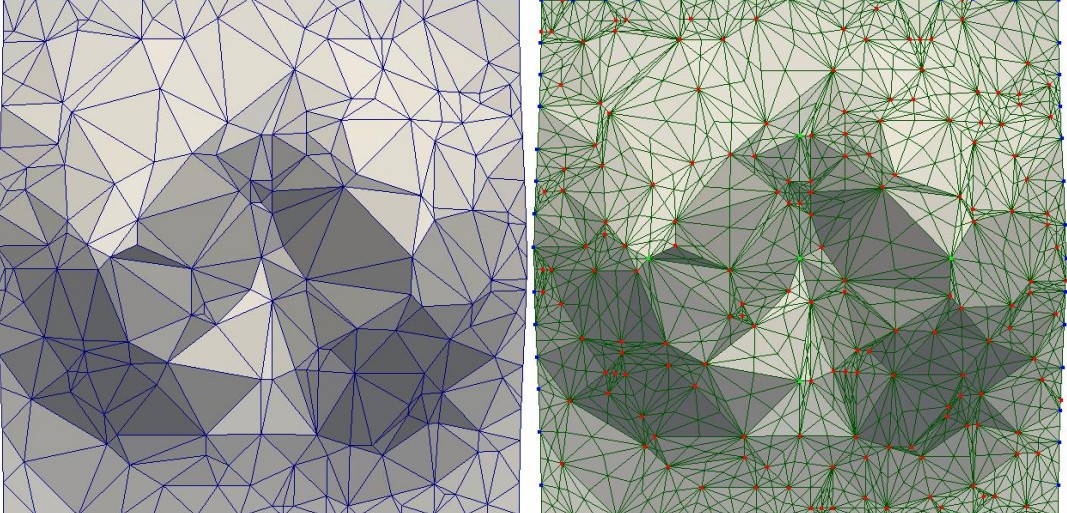

Figure 2  Initial approximate *TIN* mesh (left) and its dual mesh *TD* (right). The input sample points on the dual mesh are also rendered in red.





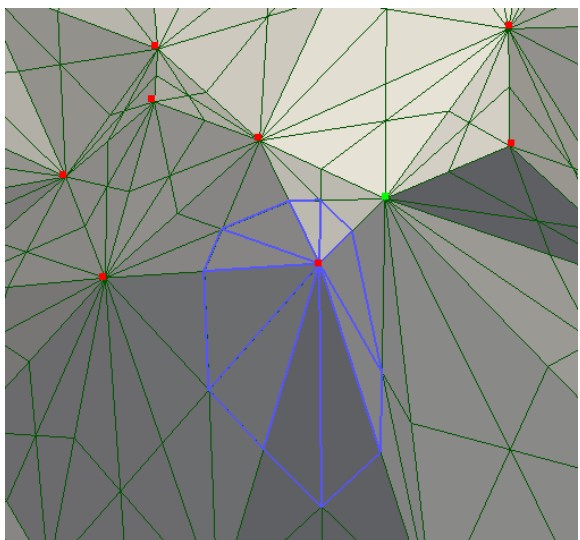

Figure 3 The incident triangles toward the first vertex $r_i$ on the dual mesh comprise an initial approximate Voronoi cell (rendered as a blue wireframe); the centre vertex is rendered in red.

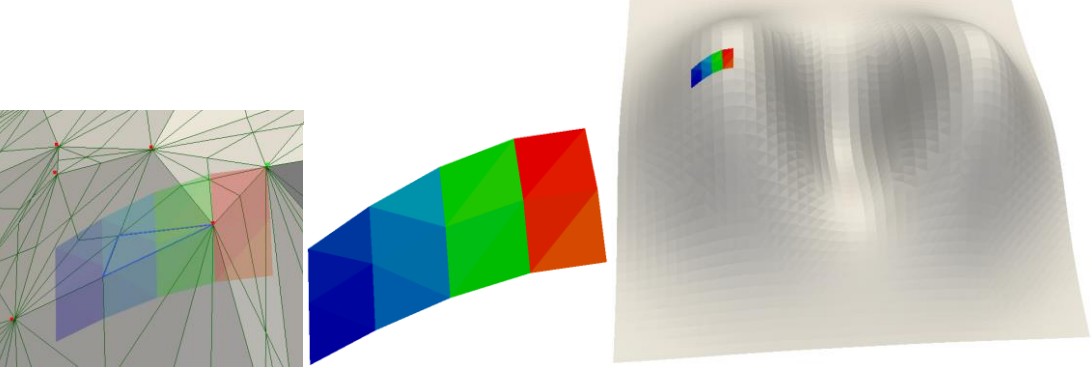

5     Figure 4 A triangle $T_j$ (rendered in semi-transparent blue) with its minimal bounding box's intersection part *narrPd* with the original mesh on the approximate mesh (left), the localized intersection part *narrPd* alongside (middle), and the intersection part on the original mesh *oriPd* (right).

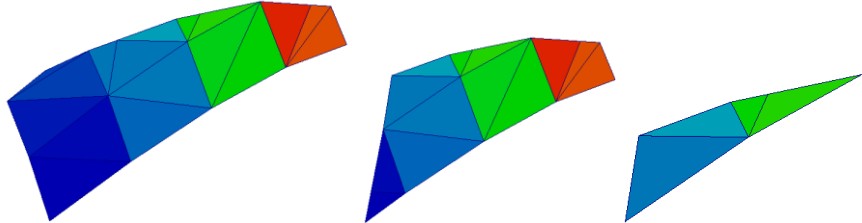

Figure 5 Exact clipping steps of *narrPd* with $T_i$. The sequence from left to right illustrates the edge clipped results. This clipping used a
10     linear interpolation to guarantee accurate energy estimation and avoid zigzagging Voronoi cells.




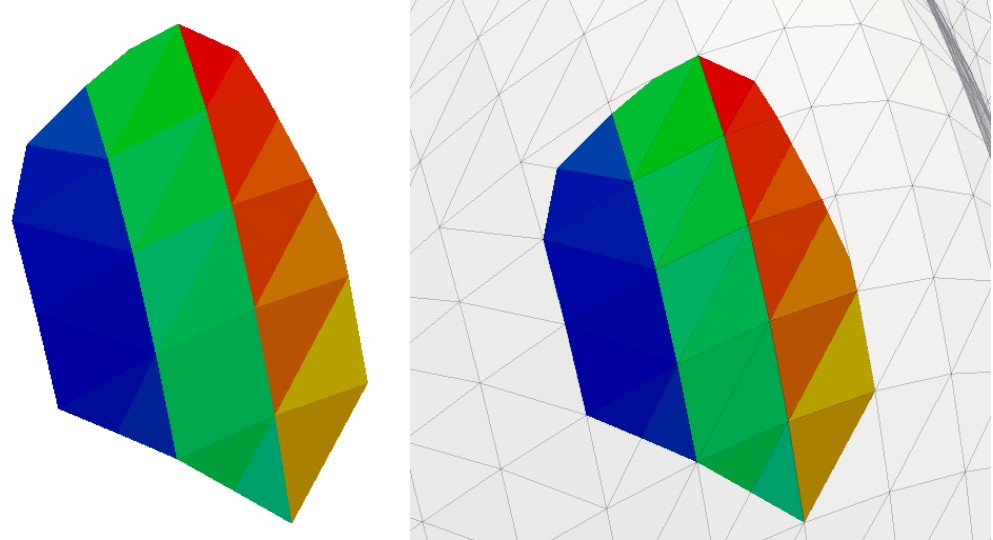

Figure 6  Reference patch $ref_j$ on the high-resolution DEM surface of an initial Voronoi cell with the centre at $r_i$  (left). Right: $ref_j$ on the original DEM surface $oriPd$.

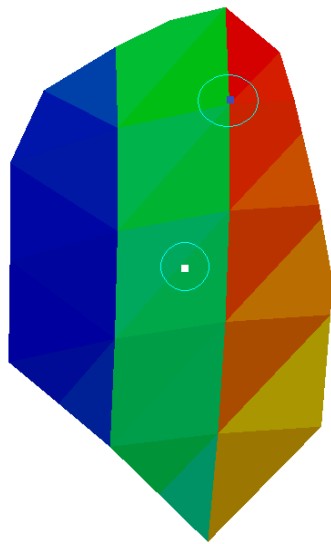

5    Figure 7  Barycentre computation based on the reference patch $ref_j$;  the original site is the white block in the circle, while the newly generated site and its projection on $oriPd$ are denoted as blue blocks in the circle.





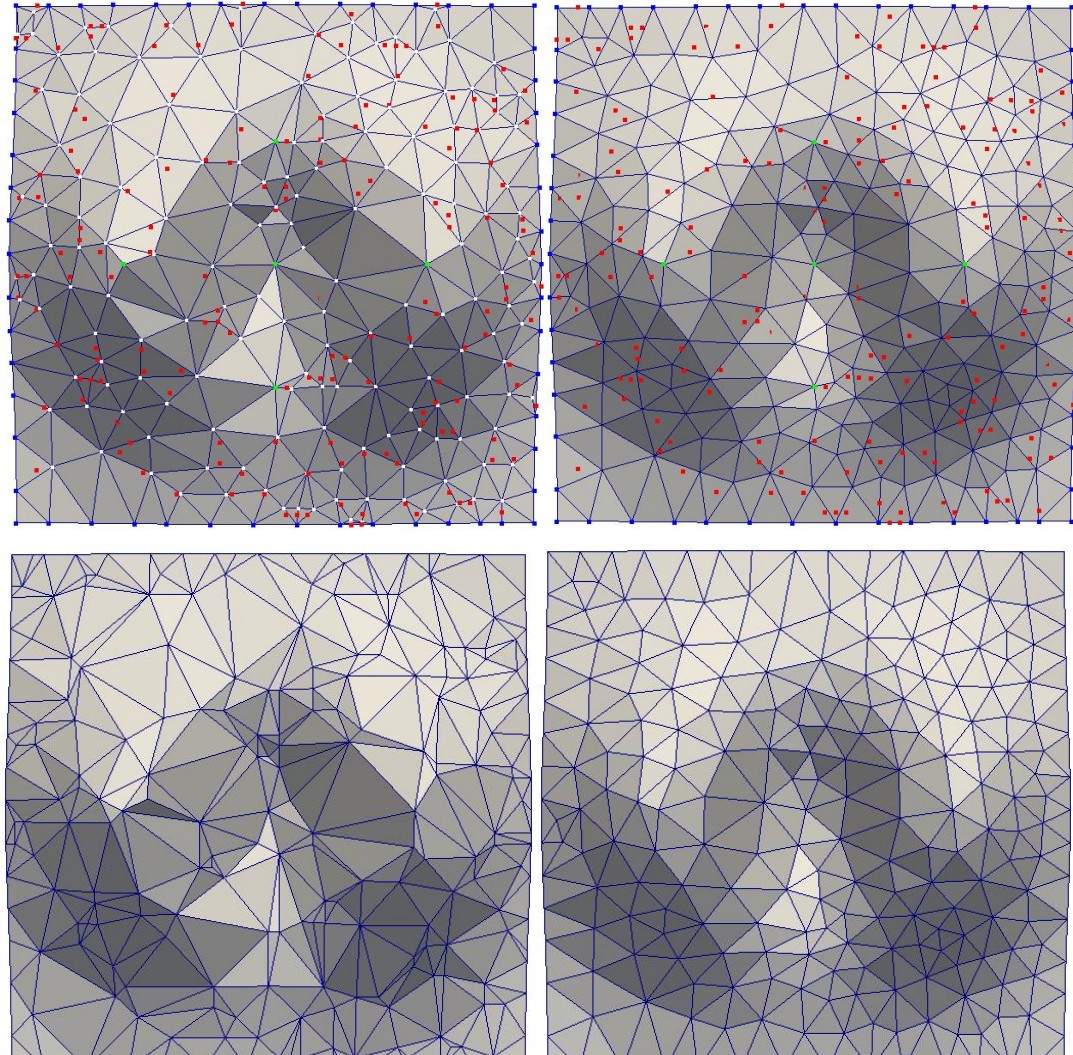





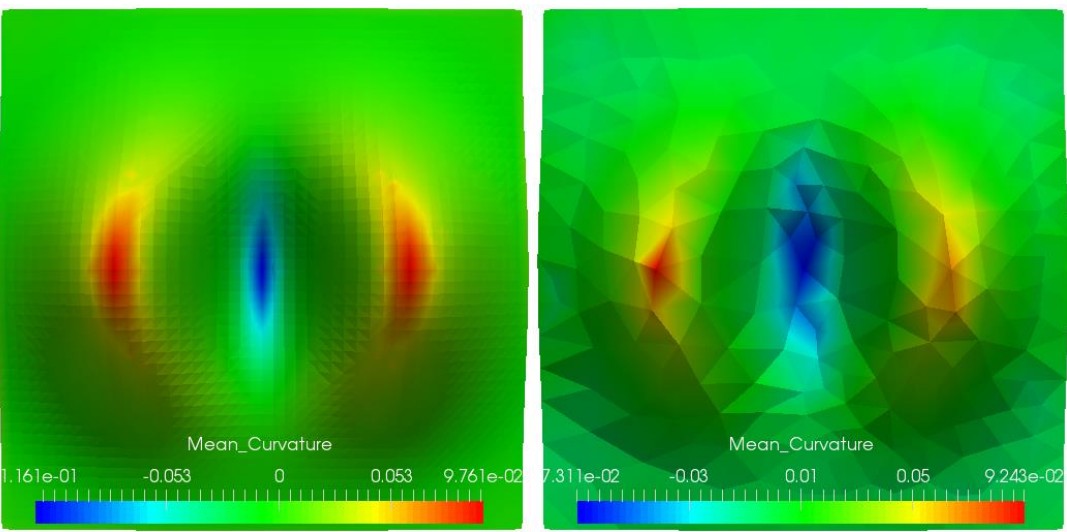

Figure 8 Converged result comparison. Top left: reconstructed TIN mesh from one iteration with the initial point sites presented. Top right: after about 140 steps, the iteration converged to a DEM equilibrium; the reconstructed TIN mesh with the initial sites is also presented. Middle: the initially approximated TIN mesh (left) and the final TIN mesh (right). Bottom: curvature distribution on the original mesh (left) and the generalized mesh (right).

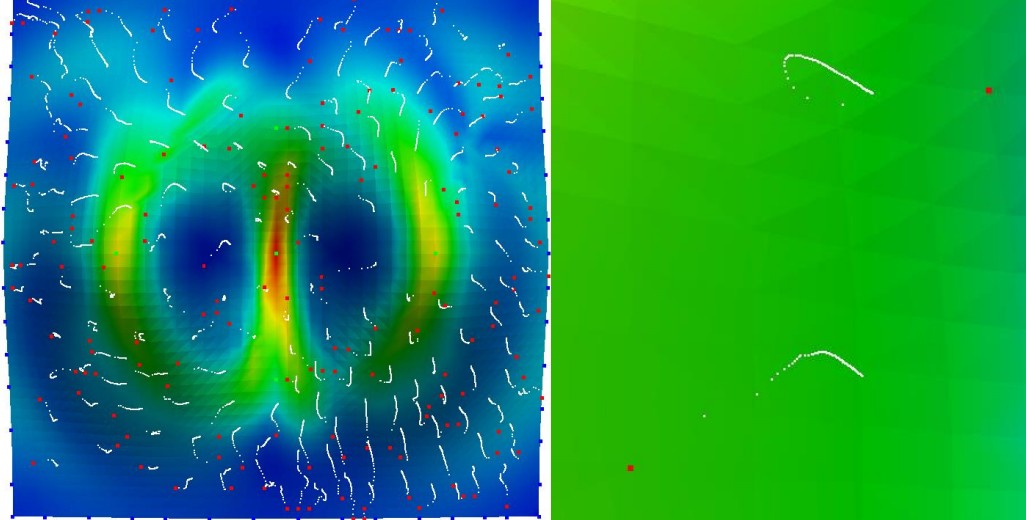

Figure 9 Trajectories of site points' convergence. The red points indicate the initial sample set, and the trajectories show the convergence trends, with closer gaps between candidate points. The right side shows a magnified view of the convergence of two points. These trends imply that the cCVT's convergence complies with *Lloyd Relaxation* linear convergence.





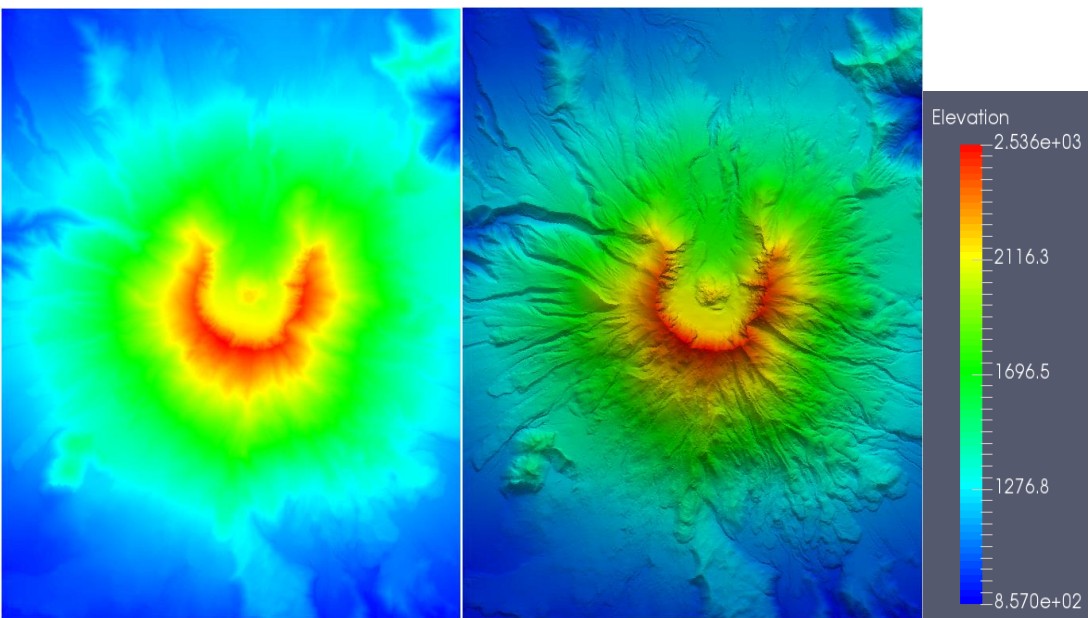

Figure 10  Experimental Site 1: Mount St. Helens. Left: image view. Middle: hillshade view. The source data were selected from a 2924x3894 grid with a 3 m cell size.

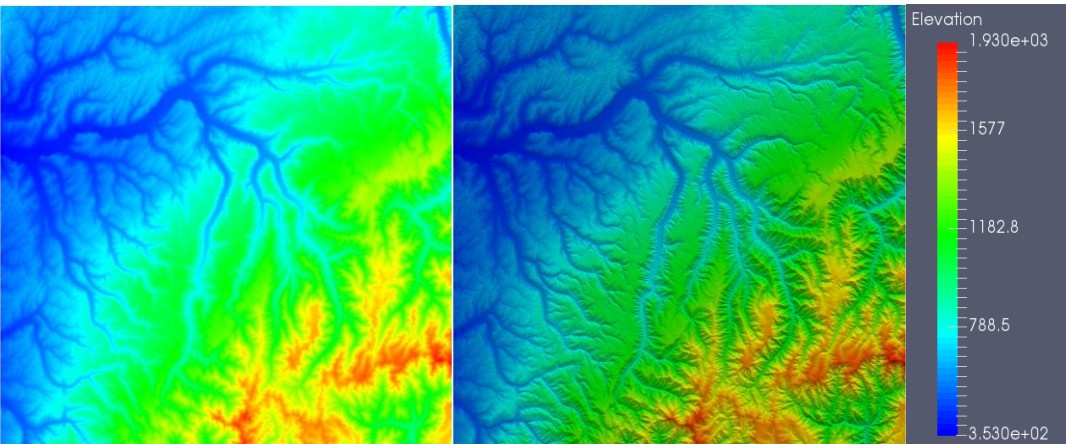

5  Figure 11 Experimental Site 2: UTM11 Zone. Left: image view. Middle: hillshade view. The source data were selected from a 3875 x 3758 grid with a 10 m cell size.

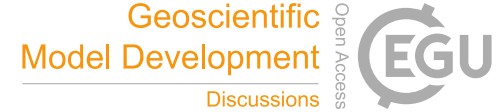



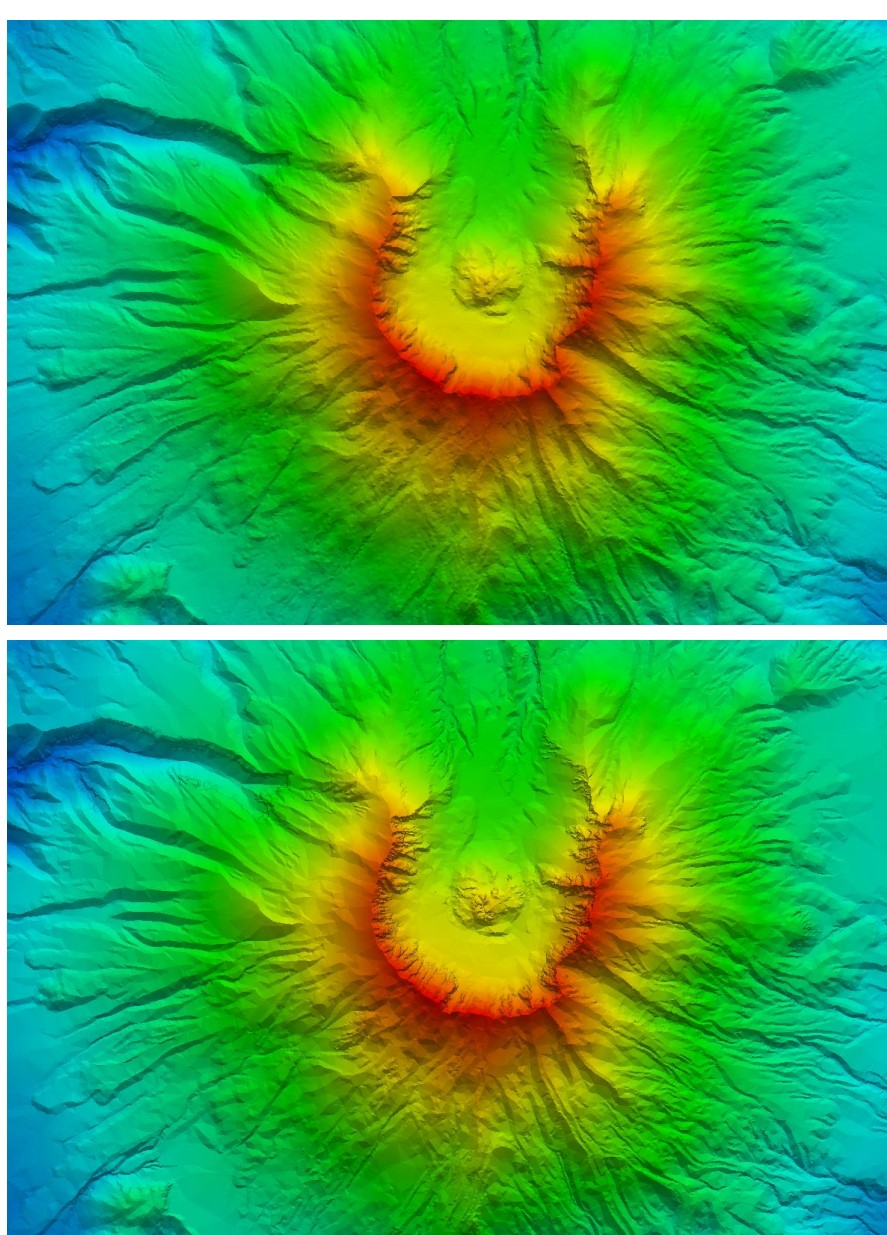

Figure 12  Visual examination of St. Helens. Top: cCVT mesh. Bottom: HFPR mesh. The latter mesh appears more rigid than the former, which implies a stronger generalisation effect and the possible loss of detail under the same transformation scale.





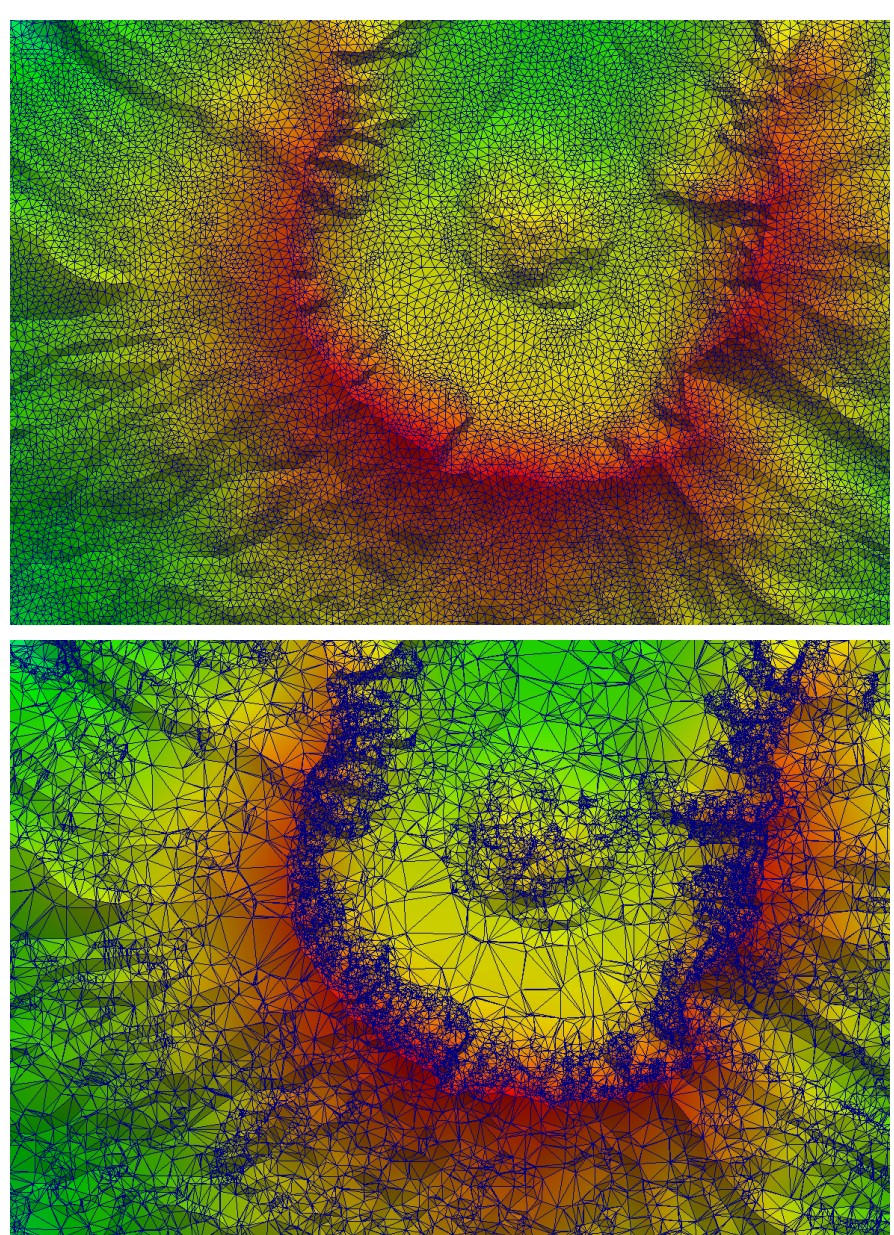

Figure 13 Mesh quality from an intuitive comparison. Top: cCVT-generalized mesh with nearly uniform triangles. Bottom: HPFR generalized mesh with irregular triangles.





Figure 14 Detail loss of the HPFR generalisation method. The inspected area in the experimental site, which is bounded by the white rectangle (a); the magnified inspection area on the original dense TIN (b); the area on the cCVT-generalized TIN (c); and the area on the HPFR-generalized TIN (d). The HPFR method generated a rougher mesh than the CVTs. Thus, structural terrain feature distortion or misconfiguration might be introduced by a stronger generalisation effect.





Figure 15  Detail loss from the HPFR method in the UTM11 dataset. (a) The inspection area in the entire experimental site. The magnified view of the inspection area is shown (b) on the original dense DEM, (c) on the cCVT-generalized TIN, and (d) on the HPFR-generated TIN mesh. The surface fold details in the white ellipse were recovered by the cCVT method but not by the HPFR method.





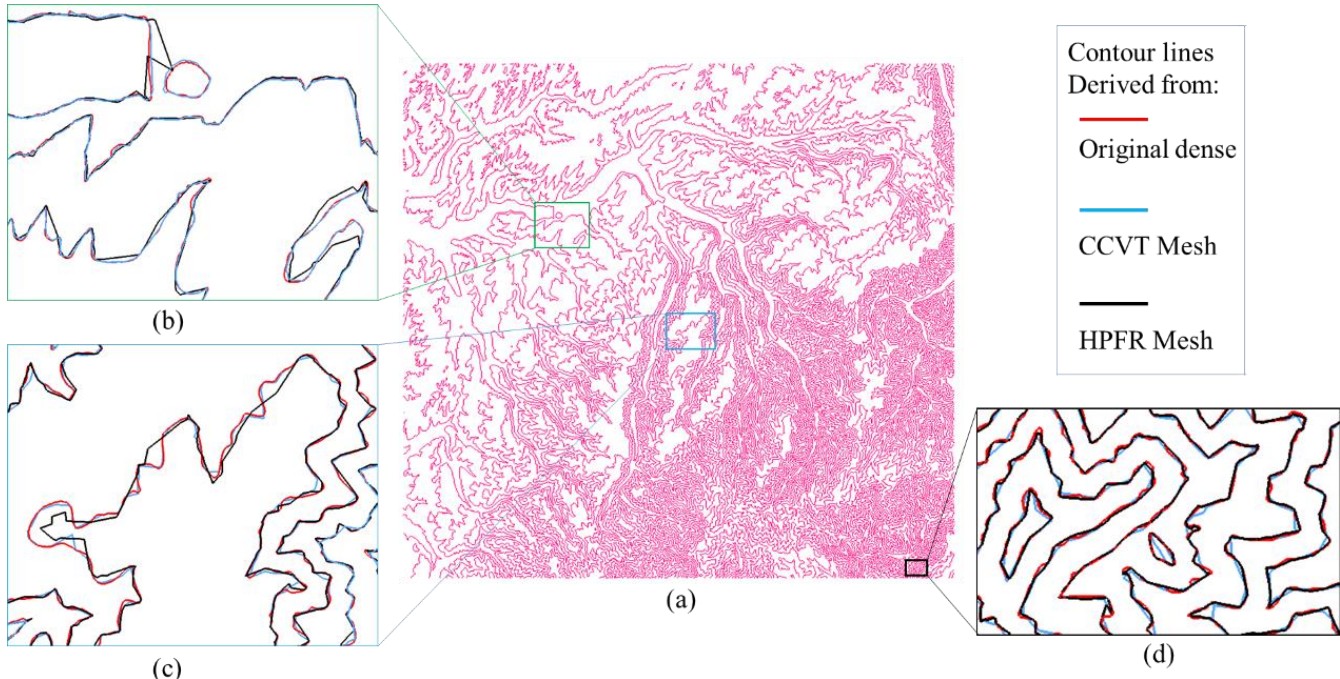

Figure 16 Comparison of the derived contour lines. The spatial distribution of the contours for the UTM11 dataset is shown in (a). The contours are spaced in 80 m increments. The contours are more densely clustered at higher elevations and under-developed areas than those in relatively low elevation area. The areas in the three boxes were magnified to show the differences in the contour configuration. The red contour lines are from the original dense DEM surface, the blue contour lines are from the cCVT-generalized TIN surface, and the black lines are from the HPFR-generated mesh. According to the illustrations in (a) and (b), the cCVT-generalized approximation generally was more accurate than the HPFR-generated approximation; the contour lines that were derived from the cCVT mesh (blue lines) are more in line with those from the original mesh. The contours that were generated by the HPFR mesh sometimes edge out on steep slopes, as shown in (d), because these areas accumulated a relatively large amount of sample points.