# Peer review of "A high-fidelity multiresolution DEM model for Earth systems"

_Geoscientific Model Development, 2016_

## Referee Comment (RC1) · Anonymous Referee #1 · 2 Sep 2016

This paper present a method for generating DEM's of differing resolution from point cloud data such as that obtained from LiDAR. Such a process is widely undertaken across the geosciences and therefore should be of interest to a range of researchers who require topography. The authors are clearly experts in their field and make a number of really nice augments for their method, however as geoscientist I found some of the introductory section difficult to follow. I'd strongly recommend the authors give substantial consideration to making the introduction more accessible to a general audience and in the process explaining or removing much of the technical jargon. For example, I found this sentence really confusing, after several attempts to read it I'm still only partially understand what is being conveyed. "With the success of Earth and environment systems in conforming theories, explaining observations with these scale diversity processes, there exists persistent demands for extending their utility into new

and expanding scopes" Perhaps you are trying to say too much in one sentence. also "to simulate land-atmosphere, land-ocean, or land-hydrology interactions, or how to perform commensurate scale transformation to the topography itself taking care of the coupling endogenous features have proven to be a quite difficult task". For me this is unnecessarily complicated way of saying this and I don't know what you mean by coupling endogenous features. I do not doubt that the argument is correct but I just feel you will alienate some readers by presenting what are quite simple concepts in a difficult to follow manner. Most importantly the validation of the technique has some import limitations that could be easily overcome. The article strongly advocates the method for the generation of multiresolution DEM's. But this aspect is never tested in relation to the HFPR with only one number of points tested. Assuming its computationally feasible I would compute the surface accuracy statistics across a wide range of resolutions and point densities. These could be added to Table 1 or as a plot. Finally, I have three questions: 1) How does the selection of feature points effect the results (e.g. what if this was done poorly) 2) What can cCVT not be compared to previous CVT methods such as that in Du et al., 2010? 3) How would the method compare to an optimal estimator such as kriging? Overall I think the paper is very interesting, however it's very difficult to follow in places and the validation of the method is quite limited.

Specific points: Introduction increasingly improved resolution –> finer resolution P3 L 13 "for clearance of confusions and distractions with terrain generalisation" is this bit necessary? I'm not sure what it means or adds to this sentence. Section 1.3: For a general geoscience audience I think there is quite a lot of jargon here. For example, what are field control points, ground checking points? What is "an exact geometry clipping-based energy estimation"? As section two goes into much detail perhaps section 1 could be written for a general audience and no presuppose a detailed understanding of CVT. Finally, this section should clearly point out what component of the proposed method is novel, it not clear if the cCVT is novel or the application to a DEM is novel or just the software is new. Section 2.3: "For example, P. J. Kennelly pointed out that,

compared to hydrologic model, curvature-generated drainage networks has more nature performance, has not limit to one single pixel thickness, has no requirement on flat filling (depressions are rather useful sometime such as flash flood modelling), and capable of delineating both convergent flow and divergent flow (Kennelly, 2008)." I don't understand this sentence, what is a hydrologic model in this context? Do you mean that for the purpose of hydrological modelling curvature-generated drainage networks are better than some other methods. Also what does more natural performance mean in this context, could you not just say it is not limited to one single pixel thickness and has no requirement on flat filling. "The localization makes geometrical operation costs minimized..." -> The localization reduces geometrical operation costs such that the efficiency of the cCVT approximation as a whole is comparable to that of the elegant clustering approach Section 3.3: To what extent is the accuracy effected by the scale transformation ratio of the HFPR method. Am I correct in thinking the number of point used by the mesh is determined by this HFPR criteria and thus would you not want to prove that the cCVT is superior across multiple resolutions (e.g. 0.5% point left, 10 % point left etc.) P11 L 5: "more natural transition effect" I'm still not sure what more natural means in this context, A different discussion is needed here. P11 L 9: Do you mean precision or accuracy of the general approximation?
* * *

---

## Referee Comment (RC2) · Anonymous Referee #2 · 12 Sep 2016

This research presents a method for generating high quality DEMs at various resolutions, using a method that provides improved accuracy relative to commonly used heuristic methods, while retaining a relative computational advantage over smooth fitting approaches such as Kriging. The article is well written and the arguments appear to be well supported with references to literature in the field. The rational for developing a new method is clearly defined, and the topic area will be of interest to a wide audience.

My main concern with this manuscript is that it may be difficult for a general audience to follow due to the very technical language used, as well as the assumption of knowledge on behalf of the reader. For instance, section 1.3 is the first instance in which the reader is given an overview of the proposed method. In this section centroidal Voronoi tessellation, field control points, ground checking points, and clipping-based

energy estimation are mentioned, assuming the reader knows what they are. Similarly sentences such as 'CVT is driven by a robust discrete curvature as density function, based on the curvature's ability on shape characteristics capturing and shape evolution' are difficult for a non-expert to understand. In order to be accessible to the full geoscience audience, the authors may wish to add a few paragraphs throughout that are written in a less technical manner in which key principles are explained assuming no prior experience in the field.

The manuscript may also be improved by adding some additional validation. As the title highlights the method is a 'high-fidelity multiresolution DEM model' it would be nice to show how the error statistics relative to other methods change over more resolutions and DEM point densities. Also, the authors highlight that there are many approaches one might use when generating a DEM. The validation is conducted against a classic heuristic approach which is defined in the introduction as sub-optimal but computationally efficient. It is interesting that the new method is more accurate, however, it would also be interesting to know how it performs against a wider range of methods. If it is feasible to add this extra analysis it would be a good addition to the manuscript.
* * *

---

## Author Response (AR1)

**Response to Anonymous Referee#1**

We would like to thank the Referee for his/her constructive comments that concerns five major subjects and several minor issues. Through a careful study on the comments, we have made modifications accordingly. The responds to the comments and the main modifications to the paper are as following:

(Review comments are reported in red.)

**Major Comments:**

1. Making the introduction more accessible to a general audience and in the process explaining or removing much of the technical jargon.

Response:
The Introduction section is reorganized by removing some technical terms and statements are reconstructed into direct, short sentences accordingly.

2. Validation across varied resolutions.…Compute the surface accuracy statistics across a wide range of resolutions and point densities. These could be added to Table 1 or as a plot.

Response:
We have carried extra experiments on those two LiDAR derived DEMs with varied resolutions. The added resolutions are ranged from 5% to 0.1% (as ranged from 3.1% to 0.6% setting in [1]). The comparison results of the surface accuracy statistically are added to Tab.1, and we copy them here for clarification:

Tab. 1Interpolated elevation RMSEs (m) at varied scale transformation *Ratios*

| Dataset | Approx. Method | 5% | 1% | 0.5% | 0.1% |
|---------|----------------|------|------|-------|--------|
| St. Helens | cCVT | 0.636 | 1.614 | 2.455 | 5.772 |
| | HFPR | 1.028 | 2.371 | 4.006 | 11.779 |
| UTM11 | cCVT | 1.239 | 3.773 | 6.593 | 19.997 |
| | HFPR | 3.087 | 6.712 | 10.137 | 28.460 |

From the results we could see that, under the same resolution (point density), transformed DEM surface from cCVT method is generally more precise than that from HFPR method. While all surface approximation precision (compared to the original) decrease as the resolution coarsened. We have added these modifications to the manuscript (P10, L23-24).

For further analysis of to what extent (though roughly) cCVT could be comparable to existing models, cCVT-based terrain adaptive grid models (TAM) with varied resolutions are subjected to fixed-resolution grid models for comparison. This experiment is taken out on flood inundation simulation, where topography condition dominates the well-known shallow-water process. Lots of methods have been proposed in this domain to generate terrain-following computational grids, here we select two most classical grid (mesh) models for comparison, that is, block-structured mesh (BM) [2] and transfinite interpolation mesh (TIM) [3].

The selected topography comes from the Okushiri tsunami experiment (c.f. Fig. 1). The BM grid (c.f. Fig. 2 a) and TIM grid (c.f. Fig.2 b) for this experiment come from ANUGA[1] validation case and TELEMAC[2] validation case respectively, both are publicly available from their official websites. TAM grid from cCVT is illustrated as Fig. 2 (c). For rough analysis, we build TAM grids with varied resolutions ranged from 24K triangles to 7.8K triangles, and compute different surface approximation metrics for comparison with the fixed-resolution BM grid (with 21K triangles) and TIM grid (with 25K triangles). The result are listed in Tab. 2.

[Figure]

Fig. 1 Topography of Okushiri Tsunami experiment. Left, elevation rendering; Right, mean curvature rendering. (CH-B, CH-5-7-9 marks the four gauge locations)

[1] ANUGA is a general-purposed hydrodynamic modelling tool developed by Australia National University and Geoscience Australia, https://anuga.anu.edu.au/.
[2] TELEMAC is an integrated solver suite for free surface flow, http://www.opentelemac.org/

[Figure]

Fig. 2 Different Computational grid models. (a) Block-structured grid, (b) Transfinite interpolation grid, (c) Terrain adaptive grid.

From Tab. 2, we can see that, under the same resolution, for any approximation metrics as Hausdorff distance, barycenter elevation interpolation (which is commonly adopted by finite volume methods), or random elevation interpolation, TAM grid (A0) approximates the original terrain surface best. TAM grid with only half samples (TAM A1) to that of BM grid or TIM grid performs approximation fairly well to that of the two comparing grids.

**Tab.2 Approximation precision comparison for different grids.**

| Approx. Metric | BM, 21K | TIM, 25K | TAM A0, 24K | TAM A1, 12K | TAM A2, 7.8K |
|---|---|---|---|---|---|
| Hausdorff Dist.(1e-2) | 4.147 | 1.205 | 0.304 | 0.354 | 1.400 |
| Bary RMSE(1e-4) | 4.127 | 2.414 | 1.653 | 2.315 | 3.035 |
| Rand RMSE(1e-4) | 3.942 | 2.823 | 2.024 | 2.844 | 3.947 |

3. How does the selection of feature points effect the results? (e.g. what if this was done poorly)

Response:
CVT works under the variational framework. The result of CVT optimization relies on initial conditions and boundary conditions. For DEM transformation, the effects of selected feature points can be summarized from two aspects:

(1) In general, if the feature points are not well selected for initial samples, we can still get improved surface approximation precision. But the feature points (critical points as well) may not be accurately positioned, this means kinds of structural distortion. The result thus may be acceptable from surface approximation precision expectation, but may not be acceptable from feature retention aspect.

(2) If the initial samples are extremely ill-positioned, CVT may fail to recover good surface approximation either. To illustrate this, we built feature points based cCVT and random points based cCVT on the former Okushiri topography (the resolution is set at 1% point density). The result surfaces are illustrated as Fig. 3 and Fig.4. The Hausdorff distance and barycenter interpolation RMSEs for the two surface are listed in Tab. 3.

From Fig.3, Fig.4, and the comparison results of Tab. 3, we can see that, though both approaches show good structural feature capturing capability, however the ill-positioned samples (i.e., weak feature capturing capability) affects the surface precision and sample points distribution greatly. As regards to the negative indices, surface approximation precision from Hausdorff distance is more evident than that from barycenter elevation interpolation RMSEs (computed independently).

These facts imply two important issues: (1) CVT might be used for surface feature extraction, i.e., terrain generalisation purpose; (2) Feature points based approach is essential to the cCVT implementation, auxiliary input points or computed structures are still useful for DEM transformation.

We have add some modifications to the manuscript to stress these points. (Section 2.3.2)

[Figure]

Fig. 3 Feature points as samples and converged distribution

[Figure]

Fig. 4 Random points as initial samples and converged distribution

**Tab.3 Surface approximation comparison for feature points based CVT and random points based CVT.**

| Approx. Metric | Feature Points CVT, 2K | Random Points CVT, 2K |
| --- | --- | --- |
| Hausdorff Dist.(1e-2) | 1.216 | 2.392 |
| Bary RMSE (1e-3) | 1.220 | 1.322 |

4. What can cCVT not be compared to previousCVT methods.

Response:
cCVT aims at the intrinsic properties of the terrain surface geometry, while CVT could be applied to a wide variety of application domains, even not confined to geometry space.

For the implementation of the proposed cCVT, we develop an extra energy referring through exact geometry clipping technique. The exact clipping is due to several considerations such as numeric instabilities, fast convergence, and quality grid.

The keys to the popular CVT implementation is to cluster facets without surface reconstruction, and it relies on existing vertices rather than generating new ones. This may result in bad grid quality as exemplified by Fig. 5 (b), which is generated by the clustering approach. This grid can be further optimized by cCVT method and the result is shown as Fig.5 (c), from which we can observe improved grid quality with smooth transition.

The exact geometric clipping functions on the presumption of the 2.5 dimensionality of DEM surface. In such circumstance, it cannot be applied to arbitrary geometric domains as clustering CVTs can.

[Figure]

Fig. 5Grid quality comparison of classical CVT clustering (b) and exact energy referring (c). They are subtracted from the former Okushiri topography (a) example.

5. How would the method compare to an optimal estimator such as kriging?

Response:
Both the Kriging method and the cCVT method consider the samples' impact from either spatial domain or frequency domain. But Kriging method is usually utilized for the situation of data scarcity and it is essentially an interpolation approach, while cCVT method considers the data redundancy problem and it is thus usually a coarsening approach.

**Specific points:**

1. increasingly improved resolution (P2L9)

Response:
We have replaced it with "finer resolution".

2. for clearance of confusions and distractions (P3L13)

Response:
We modified this sentence as:

"As surface approximation precision and terrain feature retention are competitive for the redistribution of feature points, DEM (digital elevation model) generalisation is differentiated from terrain generalisation for its emphasis on surface approximation as a whole, with the aim of providing precise surface interpolation (Guilbert et al., 2014)."

3. Section 1.3: For a general geoscience audience

Response:
The Section 1.3 is reconstructed to state aims and contributions. The involved technical considerations, underlying principles of CVT are rearranged into section 2.3, where three new sub-sections are added, for a clearer statements.

4. Section 2.3, hydrological model, curvature-generated drainage networks (P7 L22-25)

Response:
Thanks for the Referee's reminding, we modified the confusing "hydrological model" as the commonly used "flow accumulation model", and the statements is reorganized as:

"compared to the results of the flow accumulation model, curvature based delineation of drainage networks has not limited to one pixel thickness and requires no depression filling (Kennelly, 2008)".

5. The localization makes geometrical operation costs minimized (P7L32)

Response:
This section has been reconstructed into three new sub-sections for a clearer statements. The related statements here are rearranged as:

"… By this exactly clipped referring patch we compute accurate energy estimation for new approximated sites. The global clipping computation is localized using a *kd-tree* structure.

The localization and accurate referring energy computation makes cCVT iteration converge fast. The efficiency of the cCVT approximation as a whole is comparable to that of the elegant clustering approach. We go no further for the complexity analysis but however provide an implementation of the classical clustering with the same settings as the cCVT in the attachment."

6. Section 3.3: To what extent is the accuracy effected by the scale transformation ratio of the HFPR method

Response:
This has been supplied as response to the major comment#2.

7. "more natural transition effect" of cCVT optimized grid (P11 L5)

Response:
We modified the "natural transition effect" as "smooth grid transition".

We may still use the Okushiri computational meshes to explain this effect. In Fig.2 (c), the cCVT generated TAM mesh has smooth transition areas all over the domain, while block-structured mesh has abrupt transition areas. This kind of smoothness are also presented in Fig. 4 (b), compared to rigid transition grid of Fig. 4(c).

8. Do you mean precision or accuracy of the general approximation? (P11 L9)

Response:
Here we refer to approximation precision, not accuracy.

References for Response to Referee #1:
[1] Zhou, Q. and Chen, Y.: Generalization of DEM for terrain analysis using a compound method, ISPRS Journal of Photogrammetry and Remote Sensing, 66, 38-45, 2011, doi:10.1016/j.isprsjprs.2010.08.005.
[2] Nikolos, I. K. and Delis, A. I.: An unstructured node-centered finite volume scheme for shallow water flows with wet/dry fronts over complex topography, Computer Methods in Applied Mechanics and Engineering, 198, 3723-3750, 2009, doi:10.1016/j.cma.2009.08.006.
[3] Li, Y., et al.: An orthogonal terrain-following coordinate and its preliminary tests using 2-D idealized advection experiments. Geosci. Model Dev., 7(4), 1767-1778, 2014, doi: 10.5194/gmd-7-1767-2014.

**Response to Anonymous Referee#2**

We would like to thank the Referee for his/her constructive comments that concerns two major subjects. Through a careful study on the comments, we have made modifications accordingly. The responds to the comments and the main modifications to the paper are as following:

(Review comments are reported in red.)

**Referee Comments 1:**

My main concern with this manuscript is that it may be difficult for a general audience to follow due to the very technical language used, as well as the assumption of knowledge on behalf of the reader. For instance, section 1.3 is the first instance in which the reader is given an overview of the proposed method. In this section centroidal Voronoi tessellation, field control points, ground checking points, and clipping-based energy estimation are mentioned, assuming the reader knows what they are. Similarly sentences such as 'CVT is driven by a robust discrete curvature as density function, based on the curvature's ability on shape characteristics capturing and shape evolution' are difficult for a non-expert to understand. In order to be accessible to the full geoscience audience, the authors may wish to add a few paragraphs throughout that are written in a less technical manner in which key principles are explained assuming no prior experience in the field.

**Response:**

The Introduction section is reorganized by removing some technical terms and statements are reconstructed into direct, short sentences accordingly.

The Section 1.3 is reconstructed to state aims and contributions. The involved technical considerations, underlying principles of CVT are rearranged into section 2.3, where three new sub-sections are added, for a clearer statements.

**Referee Comments 2:**

The manuscript may also be improved by adding some additional validation. As the title highlights the method is a 'high-fidelity multiresolution DEM model' it would be nice to show how the error statistics relative to other methods change over more resolutions and DEM point densities. Also, the authors highlight that there are many approaches one might use when generating a DEM. The validation is conducted against a classic heuristic approach which is defined in the introduction as sub-optimal but computationally efficient. It is interesting that the new method is more accurate, however, it would also be interesting to know how it performs against a wider range of methods. If it is feasible to add this extra analysis it would be a good addition to the manuscript.

**Response:**

We have carried additional experiments for the validations, based on which some revisions are made on the manuscript. Here the more detailed explanation is outlined below.

For the error statistics comparison over more resolutions, experiments on those two LiDAR derived DEMs with varied resolutions are tested. The added resolutions are ranged from 5% to 0.1% (as ranged from 3.1% to 0.6% setting in [1]). The comparison results of the statistical surface interpolation RMSEs are added to Tab.1, and we copy them here for clarification:

Tab. 1Interpolated elevation RMSEs (m) at varied scale transformation *Ratios*

| Dataset | Approx. Method | 5% | 1% | 0.5% | 0.1% |
|---------|----------------|------|------|------|------|
| St. Helens | cCVT | 0.636 | 1.614 | 2.455 | 5.772 |
| | HFPR | 1.028 | 2.371 | 4.006 | 11.779 |
| UTM11 | cCVT | 1.239 | 3.773 | 6.593 | 19.997 |
| | HFPR | 3.087 | 6.712 | 10.137 | 28.460 |

From the results we could see that, under the same resolution (point density), transformed DEM surface from cCVT method is generally more precise than that from HFPR method. While all surface approximation precision (compared to the original) decrease as the resolution coarsened. We have added these modifications to the manuscript (P10, L23-24).

Before comparing cCVT against methods other than the heuristic approach, it might be worthwhile to note that, feature points based heuristic approaches perform DEM transformation really well than those other classical approaches such as very important point filtering (VIPs), resampling, or interpolation on neighbor grids [1, 2]. It is thus might be interesting to compare cCVT with methods come from application domains in Earth and environmental systems where topography is directly involved and topographic effects are greatly concerned.

Here we selected a block refinement grid model (BM) and a transfinite interpolation grid model (TIM) for analysis, they are two widely used computational models in the flood inundation simulation domain where topography dominates the well-known shallow-water process [3, 5]. The BM model is of preferred for its arbitrary enhancement capability [3], while TIM model is of preferred for its quality grid with smooth transition [4]. Besides the ordinary measures as averaging neighbor grid values or high-order interpolations, both models will make utilizes of their grid refinement or adaption to introduce topography variation [5, 6].

And the widely studied Okushiri tsunami experiment is taken for the inundation scenario, the topography of the Okushiri tsunami experiment is illustrated as Fig. 1. The terrain-driven [6] BM grid model (c.f. Fig. 2 a) and TIM grid model (c.f.

Fig.2 b) for this experiment come from ANUGA[3] validation case and TELEMAC[4] validation case respectively, both are publicly available from their official websites. Terrain adaptive grid model (TAM) from cCVT is illustrated as Fig. 2 (c). For rough quantitative analysis, we build TAM grids with varied resolution ranged from 24K triangles to 7.8K triangles, and compute different surface approximation metrics for comparison with the fixed-resolution BM grid (with 21K triangles) and TIM grid (with 25K triangles). The results are listed in Tab. 2.

[Figure]

Fig. 1 Topography of Okushiri Tsunami experiment. Left, elevation rendering; Right, mean curvature rendering. (CH-B, CH5-7-9 marks the four gauge locations)

[Figure]

Fig. 2 Different Computational grid. (a) Block-structured grid model, (b) Transfinite interpolation grid model, (c) Terrain adaptive grid model.
* * *
[3] ANUGA is a general-purposed hydrodynamic modelling tool, developed by Australia National University and Geoscience Australia. https://anuga.anu.edu.au/.
[4] TELEMAC is an integrated solver suite for free surface flow. http://www.opentelemac.org/

From the preliminary results in Tab. 2, we can see that, under the same resolution, for any approximation metric as Hausdorff distance, barycenter elevation interpolation (which is commonly adopted by finite volume methods), or random elevation interpolation, TAM grid (A0) approximates the original terrain surface best. TAM grid with only half samples (TAM A1) to that of BM grid or TIM grid performs fairly well to that of the two comparing grids.

**Tab.2 Approximation precision comparison for different grids.**

| Approx. Metric | BM, 21K | TIM, 25K | TAM A0, 24K | TAM A1, 12K | TAM A2, 7.8K |
|---|---|---|---|---|---|
| Hausdorff Dist.(1e-2) | 4.147 | 1.205 | 0.304 | 0.354 | 1.400 |
| Bary RMSE(1e-4) | 4.127 | 2.414 | 1.653 | 2.315 | 3.035 |
| Rand RMSE(1e-4) | 3.942 | 2.823 | 2.024 | 2.844 | 3.947 |

Deep examination of the feedbacks imposed by the improved topography representation on the hydraulic models is expected in future studies. However, for not digress from the main subject, this part of discussion along with the expected inundation study will not be added to the manuscript.

**References for Response to Referee#2:**
[1] Zhou, Q. and Chen, Y.: Generalization of DEM for terrain analysis using a compound method, ISPRS Journal of Photogrammetry and Remote Sensing, 66, 38-45, 2011, doi:10.1016/j.isprsjprs.2010.08.005.

[2] Chen, C. and Li, Y.: An orthogonal least-square-based method for DEM generalization, International Journal of Geographical Information Science, 27, 154-167, 2012, doi:10.1080/13658816.2012.674136.

[3] Nikolos, I. K. and Delis, A. I.: An unstructured node-centered finite volume scheme for shallow water flows with wet/dry fronts over complex topography, Computer Methods in Applied Mechanics and Engineering, 198, 3723-3750, 2009, doi:10.1016/j.cma.2009.08.006.

[4] Li, Y., et al.: An orthogonal terrain-following coordinate and its preliminary tests using 2-D idealized advection experiments. Geosci. Model Dev., 7(4), 1767-1778, 2014, doi: 10.5194/gmd-7-1767-2014.

[5] Bates, P.D.: Integrating remote sensing data with flood inundation models: how far have we got? Hydrological Processes, 26(16), 2515-2521, 2012.

[6] Bilskie, M.V., et al.: Terrain-driven unstructured mesh development through semi-automatic vertical feature extraction. Advances in Water Resources, 86, Part A, 102-118, 2015.

**List of changes:**

1. Introduction section, P1L28, "With the success of Earth and environment systems in conforming theories, explaining observations with these scale diversity processes, there exists persistent demands for extending their utility into new and expanding scopes (Ringler et al., 2008; Tarolli, 2014; Wilson, 2012). The pushing demand may originate from the requirements for simulating processes and scales beyond the current numerical scheme and may also emerge as response for migration from coarse gross estimation to fine regional predictions of environmental systems, as exemplified by lapse-rate controlled functional plant distributions (Ke et al., 2012), orographic forcing imposed on oceanic (Nunalee et al., 2015) and atmospheric dynamics (Brioude et al., 2012; Hughes et al., 2015), fine-grained topographic relief dominated extreme hydrological processes as flood inundations (Bilskie et al., 2015; Hunter et al., 2007), and many other geomorphological (Wilson, 2012), soil (Florinsky and Pankratov, 2015), and ecological (Leempoel et al., 2015) examples from different components of Earth systems."
was simplified as:
"With the success of Earth and environment systems with these scale diversified processes, there exists persistent demands for extending their utility into new and expanding scopes (Ringler et al., 2008; Tarolli, 2014; Wilson, 2012), as exemplified by lapse-rate controlled functional plant distributions (Ke et al., 2012), orographic forcing imposed on oceanic and atmospheric dynamics (Nunalee et al., 2015; Brioude et al., 2012; Hughes et al., 2015), topographic dominated flood inundations (Bilskie et al., 2015; Hunter et al., 2007), and many other geomorphological (Wilson, 2012), soil (Florinsky and Pankratov, 2015), and ecological (Leempoel et al., 2015) examples from Earth systems. "

2. P2L5, "how to accurately embed the underlying topography with increasingly improved resolution (with the popularity of airborne or terrestrial LiDAR technology) to simulate land-atmosphere, land-ocean, or land-hydrology interactions, or how to perform commensurate scale transformation to the topography itself taking care of the coupling endogenous features have proven to be a quite difficult task (Bilskie et al., 2015; Chen et al., 2015; Tarolli, 2014)."
was modified to
"how to accurately embed the topography with finer resolution, or how to perform commensurate scale transformation to the topography have proven to be a quite difficult task (Bilskie et al., 2015; Chen et al., 2015; Tarolli, 2014)"

3. P2L8, "Climate or weather simulation models" was changed to  "Earth and environmental simulations"

4. P2L10,  "Coupled or assimilated climate observations construct a reasonable base for dynamic or statistical downscaling, which increases the model resolvability to broader scales. However, reliant atmosphere or climate observations themselves are always of confined resolution, whilst sub-grid surfaces are designed to accommodate empirical parameterization rather

than full feature capturing, which implies bias of endogenous lateral-variability representation and mixed-up grid cell of uncertainties"

was modified as:

"Coupled or assimilated observations construct a reasonable base for dynamic or statistical downscaling. However, the observations themselves are always of confined resolution, whilst sub-grid scheme are designed for the empirical parameterization rather than intrinsic feature capturing, which implies bias of endogenous variability and mixed-up uncertainties in grid cells "

5.P2L13, "The static boundary conditions, i.e., topographic relief, are also commonly embedded through point interpolation in atmosphere-land-ocean interaction simulations, and mesh refinements are used to handle dynamic boundary conditions and minimize topographic source errors (Guba et al., 2014; Kesserwani and Liang, 2012; Nikolos and Delis, 2009; Weller et al., 2016). However, mesh from interpolated points does not necessarily comply with the terrain relief, and underlying elevation errors are frequently reported as one input uncertainty"

was changed to:

"The topography are also commonly treated as a static boundary layer in dynamics simulations, where different interpolation strategies and mesh refinement skills are used to convey terrain variation (Guba et al., 2014; Kesserwani and Liang, 2012; Nikolos and Delis, 2009; Weller et al., 2016). However, mesh from interpolated vertices does not necessarily comply with the terrain relief, and bed elevation errors are frequently reported as one input uncertainty"

6. P2L18, "While there are many situations exist where dynamic conditions are stressed for their stronger impacts on modifying prediction results than static topography conditions (Budd et al., 2015), even in the topography-driven flood processes where refinements are dynamically imposed on wet/dry fronts (Cea and Bladé, 2015; Nikolos and Delis, 2009), but the underlying topographic layer is still prominently important for its increasingly improved fidelity to the Earth's surface and vast prospective in widening application scenarios (Bates, 2012; Tarolli, 2014), and a sophisticated topography transformation treatment would be beneficial by minimizing discrepancies arisen from physical inconsistencies (Chen et al., 2015; Glover, 1999; Ringler et al., 2011)."

was changed to:

"While there are many situations where dynamic conditions are stressed for stronger impacts on modifying predictions (Cea and Bladé; Budd et al., 2015), but the underlying topography is still prominently important for its increasingly improved fidelity to the Earth's surface (Bates, 2012; Tarolli, 2014), and a sophisticated topography transformation would be beneficial to reduce discrepancies arisen from physical inconsistencies (Chen et al., 2015; Glover, 1999; Ringler et al., 2011)."

7. P2L24, "Systematic scale transformation of topographic data considering loyal fidelity has long been studied under terrain generalisation"

was changed to:

"Systematic scale transformation of topographic data has long been studied under terrain generalisation"

8. P2L33, "with the aim of providing best surface interpolation, but shares a mutual goal of detail reduction for clearance of confusions and distractions with terrain generalisation (Guilbert et al., 2014)."

was modified as:

"with the aim of providing precise surface interpolation (Guilbert et al., 2014)"

9. P3L29, the Section 1.3 was totally rearranged to state aims and contributions, as the Referees suggested.

10. P4L25, the Section 1.4, "The rest of the article is organized as follows. In Section 2 the theory behind CVT energy minimization iteration for generalized DEM surfaces is introduced, techniques for incorporating DEM generalisation principles and fast convergence are presented, and the differences between the CCVT method and classical CVT clustering approach are discussed. In Section 3, the CCVT model is evaluated with real LiDAR-derived terrain datasets to evaluate surface approximation precision and grid quality, the experimental datasets description and comparison method qualification are also presented. Section 4 discusses the CCVT's considerations, comparable results, underlying causes, and interpretations. Finally, Section 5 presents short conclusion and outlooks briefly."

was simplified as:

"The rest of the article is organized as follows. In Section 2 the theory behind CVT for optimized DEM surfaces is introduced, techniques for incorporating DEM generalisation principles and fast convergence are presented, and the differences between the cCVT implementation and classical clustering approach are discussed. In Section 3, the cCVT model is tested with real LiDAR-derived terrain datasets. Section 4 discusses some considerations, comparable results, possible causes, and interpretations of the cCVT model. Finally, Section 5 presents short conclusion and outlooks briefly."

11. P6L10, the Section 2.3 was reconstructed to accommodate added statements and added sub-sections.

12. P7L1, "Based on the above observations and requirements,"

was modified to add new comments as "Based on these observations, and consider requirements of the CVT variational framework,"

13. P7L15, "The global clustering computation is thus localized (a kd-tree is utilized for the trick) and accurate referring energy computation makes iteration converge fast. And more important, we successively approximate Voronoi tessellations

but avoid problematic clustering. Although costly geometrical operations are employed, the efficiency of the CCVT approximation as a whole is comparable to that of the elegant clustering approach, while the CCVT-generated Voronoi cells are free of zigzagging boundaries (we implement classical clustering with the same settings as the CCVT in the attachment)." was simplified as:

"The global clipping computation is localized using a kd-tree structure.

The localization and accurate referring energy computation makes cCVT iteration converge fast. The efficiency of the cCVT approximation as a whole is comparable to that of the elegant clustering approach. We go no further for the complexity analysis but however provide an implementation of the classical clustering with the same settings as the cCVT in the attachment."

14. P8L18, the phrase "generalized scale" was replaced with "transformation scale". This kind of modification were taken throughout the manuscript.

15. P9L1, "Notably, the CCVT iteration used a direct reference on the original DEM surface. The exact geometry clipping linearly interpolated the actual high-resolution surface, which guaranteed accurate energy estimation and avoided zigzagging Voronoi cells. "
was modified to stress the purpose of the proposed exact geometry clipping as:
"Notably, the direct reference on the original DEM surface is realized by the exact geometry clipping, which linearly interpolated the high-resolution surface literally. This clipping technique has several important benefits: it guarantees accurate energy estimation, it avoids the generation of invalid clustering cells or zigzagging cells, and it promises exact site position calculation which will warrants improved grid quality."

16. P10L9, new sub-sections were added for a clearer comparison. Major modification were taken here, validations with varied resolution experiments were carried on these two LiDAR-derived DEMs. Added comparison results were added to Table 1 (P15).

[revised manuscript text omitted]

批注 [o19]: Response to Referee#1, Referee#2

批注 [o20]: Response to Referee#1, Referee#2

**3.4 Qualitative comparison**

批注 [o21]: Response to Referee#1, Referee#2

[revised manuscript text omitted]

---

## Referee Report (RR1)

The authors have addressed the two main concerns raised in the original manuscript. They have done a particularly good job in making it more accessible to the scientific community through the addition of less technical elements to the manuscript.

I have only a very minor revision to suggest. There are a number of instances where sentences should be edited for clarity due to grammatical errors. A few examples include:

- P21 L9: '…that the main relief features [are] strongly stressed…'

- P21 L16-18: 'The first class… exponential time'. Restructure this sentence.

- P22 L23: 'It [is] thus a [totally] different approach…'

- P25 L9: 'Based on these observations, and [considering the] requirements of the CVT…'

---

## Author Response (AR2)

**Response to Editor review comments**

We would like to thank Topical Editor Dr. Jeffrey Neal for his careful work to the manuscript, and we especially appreciate his insightful comments that concerns presentation and language issues. We have carefully read all the comments and made modifications accordingly. Besides presentation modifications we have made, the revised manuscript is also sent out to a professional language service for polishing. For the code availability section, we have reconstructed it for a clearer description. The responds to the comments and the main modifications to the paper are as following:
(Review comments are reported in red.)

**[Specific comments]**

Abstract

1. "The topographic impacts on modifying Earth systems variability have been well recognised." This would be simpler to understand as "The impact of topography on Earth systems variability is well recognised"

**Response:**
We have modified it as the suggestion:
"The impact of topography on Earth systems variability is well recognised. "

2. This sentence "Numerical schemes from Earth systems either use empirical parameterization as sub-grid scale and downscaling skills to express topographic endogenous processes, or rely on insecure point interpolation to induce topographic forcing, which create bias and input uncertainties" should perhaps read "Numerical schemes of Earth systems either use empirical parameterization at sub-grid scale with downscaling to express topographic endogenous processes, or rely on insecure point interpolation to induce topographic forcing, which create bias and input uncertainties"

**Response:**
We have modified it as the suggestion:
"Numerical schemes of Earth systems either use empirical parameterization at sub-grid scale with downscaling to express topographic endogenous processes, or rely on insecure point interpolation to induce topographic forcing, which create bias and input uncertainties."

3. "This article proposes a novel high-fidelity multiresolution DEM model with high-quality grids to meet the challenges of scale transformation." I don't understand the "with high quality grids". Do you mean for generalising high quality topographic data?

**Response:**

We are sorry for the improper presentation, the phrase "with high quality grids" should be written as "with guaranteed grid quality". The complete context is modified as:

"DEM (digital elevation model) generalisation provides more sophisticated systematic topographic transformation, but existing methods are often difficult to be incorporated because of unwarranted grid quality. Meanwhile, approaches over discrete sets often employ heuristic approximating which are generally not best performed. Based on DEM generalisation, this article proposes a high-fidelity multiresolution DEM model with guaranteed grid quality for Earth systems."

4. Change "The cCVT model is then evaluated on real LiDAR-derived DEM datasets compared to the classical heuristic method." to "The cCVT model was evaluated on real LiDAR-derived DEM datasets and compared to the classical heuristic method."

**Response:**
We have changed it as the suggestion:
"The cCVT model was evaluated on real LiDAR-derived DEM datasets and compared to the classical heuristic method."

**Introduction**

5. "However, as numerical simulation systems evolved to incorporate broader scales and finer processes to produce more fidelity predictions" I would consider using the wording more exact predictions rather than more fidelity predictions.

**Response:**
The sentence is changed as the suggestion:
"However, as numerical simulation systems evolved to incorporate broader scales and finer processes to produce more exact predictions."

6. "Earth and environmental simulations usually adopt sub-grid scheme to exert topographical heterogeneity and rely on downscaling the finer observations to surface variables" Do you mean upscaling rather than downscaling. e.g. upscaling fine topographic observations to model surface variables. You meaning here has implication for the following sentences also.

**Response:**
We are sorry for the in-proper implications here. We do not mean to use upscaling instead of downscaling, however, we want to argue that topography via sophisticated transformation might be useful to reduce simulation bias and uncertainties. For not mislead/dissuade readers, we have rewritten this sentence and the following sentence as:
"Earth and environmental simulations usually adopt sub-grid schemes to express topography heterogeneous processes (Fiddes and Gruber, 2014; Kumar et al., 2012; Wilby and Wigley, 1997). The sub-grid schemes are designed for the empirical parameterisation rather than accurate topography representation, which often leads to mixed-up uncertainties and bias of endogenous variability (Jiménez and Dudhia, 2013; Nunalee et al., 2015). However, under-resolved representation could be improved by

variable-resolution enhancement, and bias of simulations can be justified by more fidelity topography transformation (Nunalee et al., 2015; Ringler et al., 2011)."

7. "The topography are also commonly treated as a static boundary layer in dynamics simulations" should read "Topography is commonly treated as a static boundary layer in dynamics simulations"

**Response:**
We have modified it as the suggestion:
"Topography is commonly treated as a static boundary layer in dynamics simulations"

8. "However, mesh from interpolated vertices does not necessarily comply with the terrain relief, and bed elevation errors are frequently reported as one input uncertainty" -> "However, a mesh from interpolated vertices does not necessarily comply with the terrain relief, and elevation errors are frequently reported as one input uncertainty"

**Response:**
The sentence is modified according to the suggestion:
"However, a mesh from interpolated vertices does not necessarily comply with the terrain relief, and elevation errors are frequently reported as one input uncertainty"

9. "While there are many situations where dynamic conditions are stressed for stronger impacts on modifying predictions" do you mean something like "Although there are many situations where dynamic conditions have stronger impacts on modifying predictions"

**Response:**
Yes, we mean "Althgough" instead of "While", we modified it as:
"Although there are many situations where dynamic conditions have stronger impacts on modifying predictions"

Section 1.2

10. "with the effects that the main relief features strongly stressed while non-structural details are massively suppressed" - > "with the effects that the main relief features are strongly stressed while non-structural details are massively suppressed"

**Response:**
We have modified the sentence as the suggestion:
"with the effects that the main relief features are strongly stressed while non-structural details are massively suppressed"

11. "map cognitive efforts are drawn to produce progressive data reduction" Sorry but it is not clear to me what is meant by map cognitive efforts are drawn

**Response:**

Here we would like to give some explanation. Some researchers, Ai and Li (2010), for example, had employed generalisation operations under map generalisation guidance to do DEM generalisation. One important map generalisation paradigm can be exampled as the extraction of morphological lines (contours, ridges, streamlines, etc.), for these structures are most effective to "abstracting" a shape which has coherence to human (or map) cognition. Map generalisation tools has evolved a many of generalisation operators, such as smoothing, simplifying, deforming, etc.
We have modified this sentence as:
"Terrain generalisation emphasises geomorphology or landform depiction, where map generalisation measures (such as abstracting, smoothing) are drawn to produce progressive data reduction."

12. "Since the static topographic layer are commonly composed directly by DEM datasets to diverse simulation interests, maintaining precise surface approximation 5 for rigorous boundary conditions is more important than 'sparse' geomorphology representation." - > "Since static topographic layers are commonly composed directly from DEM datasets for diverse simulation interests, maintaining precise surface approximation for rigorous boundary conditions is often more important than 'sparse' geomorphology representation."

**Response:**
This sentence has been modified as the suggestion:
"Since static topographic layers are commonly composed directly from DEM datasets for diverse simulation interests, maintaining precise surface approximation for rigorous boundary conditions is often more important than 'sparse' geomorphology representation."

13. "The first class of approaches is due to the computational bottleneck consideration, that determination which combination of vertices to a TIN surface approximates the original dense DEM surface best needs exponential time" This sentence needs to be rewritten, the meaning is unclear.

**Response:**
This sentence has been rewritten for a clearer statement as:
"The first class of approaches is due to the computational feasibility consideration, for selecting a TIN surface that best approximates the original DEM surface from exhaustively enumerating (of triangular combinations) requires exponential time (Chen and Li, 2012; Heckbert and Garland, 1997)."

14. Avoid the use of non-scientific terms like greedy, weeding

**Response:**
We removed the "greedy" term, for the involved sentence has contained enough information for the algorithm.
For the term "weeding" we replaced it with "rejection". Here we explain it a little more: When selecting feature points (critical points, salient points, or significant points) from discrete sets, there may exists too much candidates due to numerical nature. Slight numeric difference would result in a feature point. DEM generalisation over discrete set often need some kind of "filtering" mechanism to get rid of the redundant feature points caused by this numeric sensitivity, Chen and Li (2012) described this process as "weeding".

15. "The second class of approach is due to the consideration of TIN surface from feature points do not necessarily warrants best approximation to the original dense DEM, for feature points are commonly selected through some local metric." I did not understand this sentence.

**Response:**

We have rewritten this sentence as:

"The second class of approaches recognised the TIN surface constructed from locally computed feature points as not-well approximation to the original DEM surface. Much research thus considered global approximation instead of relying on an elaborate feature point selection scheme."

16. " The purpose of this article is to devise a multiresolution DEM model that considers optimized surface approximation and guaranteed grid quality. The quality grid is demanded by the aforementioned easy incorporation with the simulation systems. Multiresolution is an effective paradigm to model scale diversity (Du et al., 2010; Guba et al., 2014; Ringler et al., 2011; Weller et al., 2016), among those promising plans, we are especially fascinated by the centroidal Voronoi tessellations 4 (CVTs) method for the intuitive way to redistribute samples with a designated function (Du et al., 1999; Du et al., 2010; Ringler et al., 2011), and develop CVT to an optimized surface transformation method to realize multiresolution terrain model" - > "The purpose of this article is to devise a multiresolution DEM model that optimises surface approximation and guaranteed grid quality, where the quality grid is demanded by the aforementioned easy incorporation with simulation systems. Multiresolution is an effective paradigm to model scale diversity (Du et al., 2010; Guba et al., 2014; Ringler et al., 2011; Weller et al., 2016). Amongst a number of promising approaches, we are especially fascinated by the centroidal Voronoi tessellations (CVTs) method as an intuitive way to redistribute samples with a designated function (Du et al., 1999; Du et al., 2010; Ringler et al., 2011) to develop an optimized surface transformation method to realize multiresolution terrain models."

**Response:**

Thanks to the editor, we have modified it as the suggestion:

"The purpose of this article is to devise a multiresolution DEM model that optimises surface approximation and guarantees grid quality that can be easily incorporated into the simulation systems. Multiresolution is an effective paradigm to model scale diversity (Du et al., 2010; Guba et al., 2014; Ringler et al., 2011; Weller et al., 2016). Amongst a number of promising approaches, we are especially fascinated by the centroidal Voronoi tessellations (CVTs) method as an intuitive way to redistribute samples with a designated function (Du et al., 1999; Du et al., 2010; Ringler et al., 2011) to develop an optimised surface transformation method to realise multiresolution terrain models."

17. " The selection of feature points have important morphological structures embedded, computed (such as D8 flow algorithm) or auxiliary input morphological lines have been proved to have significant influence on the quality of transformed DEM surface." I'm not sure what you mean here, should this read something like "The selection of feature points can have a significant influence on the quality of the transformed DEM surface, [meanwhile] by embedding morphological structures such as those from D8 flow algorithms and auxiliary morphological lines."

**Response:**

Here we want to explain a little more for clarification. When building generalized DEM surface, it has been proven that selecting of feature points would be the most effect way. The approximation precision of the reconstructed DEM surface could be improved by embedding auxiliary or computed structural lines, this is because these structural information might be averaged and get lost when building the regular gridded raw DEM dataset.

Based on this recognition, we have modified the statement as:

"The selection of feature points have important morphological structures embedded (in the form of

serialised points sequences), for computed (such as D8 flow algorithm) or auxiliary input morphological lines has been proved to have significant influence on the quality of the transformed DEM surface (Zhou and Chen, 2011). The proposed method keeps the structural lines in the optimisation loop and makes it different to existing CVT implementations where stationary points are commonly not considered."

18. "For the discrete TIN surface, we compute robust mean curvature as incident property. Upon this discrete spatial domain and property domain, the CVT loops and make discrete set equalized from both domains. Spatial equalization warrants a quasi-uniform quality TIN grid, while the intrinsic property domain equalization warrants distribution of the discrete facets conforms to inherent terrain variation. It thus a total different approach to DEM generalisation and we called it curvature CVT (cCVT)" I'm sorry but I don't understand this sentence.

**Response:**
Here we would explain for clarification. Basically, CVT functions in a two-step manner optimization, spatial equalization from Voronoi tessellation and frequency equalization from barycenter adjustment. This processing might be somehow analogous to bi-lateral filtering. Based on the aforementioned observation (of DEM generalisation experience) that feature points would be most effective to construct a generalized DEM surface, and curvature has the capability describing the surface morphology, we compute the discrete curvature and take it as frequency distribution. Using this developed cCVT, we can hope of redistribution of selected samples conforming to terrain variation.
Based on this recognition, we reconstruct the sentence for a clearer statement as:
"For the discrete TIN surface, we compute robust mean curvature on each facet. The attached curvature acts as a frequency distribution. In this discrete spatial domain and frequency domain, the CVT loops and makes sample facets equalised from both domains. Spatial equalisation warrants a quasi-uniform grid quality, while the curvature domain equalisation warrants adaptive distribution conforming to the terrain relief. It is thus a totally different DEM generalisation approach, and we called it curvature CVT (cCVT)."

19. "Existing CVT implementation often undertake clustering approach. However, clustering over discrete sets suffers from numeric issues such as zigzag boundaries, invalid cluster cells (Valette et al., 2008), and limited grid quality." - > Existing CVT implementations often undertake a clustering approach. However, clustering over discrete sets suffers from numeric issues such as zigzag boundaries, invalid cluster cells (Valette et al., 2008) and limited grid quality."

**Response:**
We have changed it as the suggestion:
"Existing CVT implementations often undertake a clustering approach. However, clustering over discrete sets suffers from numeric issues such as zigzag boundaries, invalid cluster cells (Valette et al., 2008) and limited grid quality."

**[Code Availability]**

Furthermore, the description of the code availability is too vague. Specifically, what are the third party components of the code, in what software was the classic CVT implementation made. What are the novel aspects of the code and what comes from a third party? The paper must comply with the GMD data policy on the link below and this section needs to be more informative than simply asking the reader to email the authors:
http://www.geoscientific-model-development.net/about/code_and_data_policy.html

**Response:**

We implemented the cCVT and the comparing classical clustering algorithm using c++, with the open source Visualisation Toolkit (VTK, https://vtk.org). VTK provides wonderful flexible manipulation for polygonal datasets on vertex and cell level, but lacks easy control on edge structure. So, for the edge bi-sectioning based Voronoi tessellation approximation (approximated dual operation), we use the implementation of Valette et al. (2008). However, all other functions such as exact clipping and energy referring are implemented using VTK.

We have reconstructed this section as:

[revised manuscript text omitted]

Response to Comment #12

22. P3L6. "two broad classes", Modified from "two broad groups"

23. P3L6. "heuristic refinements and smooth-fitting", Modified from "heuristic refinements and smooth-fittings"

24. P3L7. "The first class of approaches is due to the computational feasibility consideration, for selecting a TIN surface that best approximates the original DEM surface from exhaustively enumerating (of triangular combinations) requires exponential time (Chen and Li, 2012; Heckbert and Garland, 1997)."
Response to Comment #13

25. P3L10. "in which insertion (or deletion) refinements on feature points are adopted", Modified from "which insertion (or deletion) refinements on feature points"

26. P3L12. "prohibitive", Modified from "inhibitive"

27. P3L17. "The second class of approaches recognised the TIN surface constructed from locally computed feature points as not-well approximation to the original DEM surface. Much research thus considered global approximation instead of relying on an elaborate feature point selection scheme,"
Response to Comment #15

28. P3L23. "rejection", Response to Comment #14

29. P3L25. "well performing with respect to loyalty to the original terrain surface nor easily incorporated by the numerical schemes" , Modified from " best performed with loyalty to the original terrain surface, nor easily incorporated by the numerical schemes"

30. P3L28. "The purpose of this article is to devise a multiresolution DEM model that optimises surface approximation and guarantees grid quality that can be easily incorporated into the simulation systems. Multiresolution is an effective paradigm to model scale diversity (Du et al., 2010; Guba et al., 2014; Ringler et al., 2011; Weller et al., 2016). Amongst a number of promising approaches, we are especially fascinated by the centroidal Voronoi tessellations (CVTs) method as an intuitive way to redistribute samples with a designated function (Du et al., 1999; Du et al., 2010; Ringler et al., 2011) to develop an optimised surface transformation method to realise multiresolution terrain models."
Response to Comment #16

31. P4L2. "this general optimisation method", Modified from "this general purposed optimization method"

32. P4L3. "we made the following contributions:", Modified from "we made a few contributions as follow"

33. P4L4. "The generalised DEM surface is initially approximated by a triangular grid constructed from selected feature points. The selection of feature points have important morphological structures embedded (in the form of serialised points sequences), for computed (such as D8 flow algorithm) or auxiliary input morphological lines has been proved to have significant influence on the quality of the transformed DEM surface (Zhou and Chen, 2011). "
Response to Comment #17

34. P4L10. "For the discrete TIN surface, we compute robust mean curvature on each facet. The attached curvature acts as a frequency distribution. In this discrete spatial domain and frequency domain, the CVT loops and makes sample facets equalised from both domains. Spatial equalisation warrants a quasi-uniform grid quality, while the curvature

domain equalisation warrants adaptive distribution conforming to the terrain relief. It is thus a totally different DEM generalisation approach, and we called it curvature CVT (cCVT)"
Response to Comment #18

35. P4L15. "Existing CVT implementations often undertake a clustering approach. However, clustering over discrete sets suffers from numerical issues such as zigzag boundaries, invalid cluster cells (Valette et al., 2008) and limited grid quality. "
Response to Comment #19

36. P4L19. "has a global optimisation mechanism", Modified from "but has global optimization mechanism"

37. P4L20. "which can be used to build", Added "can be used to"

38. P4L28. "Section 5 briefly presents a short conclusion and outlook.", Edited from "Section 5 presents a short conclusion and outlooks briefly."

39. P5L10. "by summing up the potential energies of all cells Vi:", Modified from "by summing up every cell Vi's potential energy"

40. P5L12. "The energy minimiser is", Added "The"

41. P5L14. "since", Modified from "for"

42. P5L19. "The most classical energy minimisation process of centroidal Voronoi tessellation is expressed by Lloyd's Relaxation (Lloyd, 1982). The main idea of this algorithm is to first tessellate the surface, and then perform density integration over the area to find a 'gravity' barycentre for each tessellated cell, which is used as the new site for the iteration. The pseudo code of this procedure is shown below."
Correspond to Change #24

43. P6L8. "projecting", Modified from "project"

44. P6L8. "using instead the constrained projection point for the new site", Modified from "the constrained projection point is used instead"

45. P6L10. "for more accurate site calculations", Modified from "for further accurate site calculation"

46. P6L11. "Fast convergence to DEM equilibria", Modified from "Fast converge to DEM equilibrium"

47. P6L22. "zigzag cluster cell boundaries", Edited from "zigzag cluster boundaries"

48. P6L24. "Furthermore", Added for smoother statement

49. P6L29. "derivatives", Modified from "derivative"

50. P6L29. "will describe a more fundamental terrain geometry shape", Edited from "will be used to describe a more

detailed set of terrain surface parameters"

51. P7L1. "considering", Modified from "consider"

52. P7L5. "Curvature's ability to flexibly describe terrain morphology", Edited from "Curvature's flexible ability on depicting terrain morphology"

53. P7L6. "noted", Modified from "pointed out"

54. P7L6. "compared to the results of the", Modified from "compared to the result of"

55. P7L7. "is not limited to", Modified from "has not limit to"

56. P7L11. "an edge bi-sectioning based dual operation", Added for clearer statement

57. P7L19. "that of the elegant clustering approach (also has kd-tree fast location embedded)", Added statement for clarification

58. P8L18. "For effectiveness", Modified from "As for effectiveness"

59. P8L19. "set the transformation scale to", Modified from "set the transformation scale at"

60. P8L19. "there are approximately", Modified from "there are about approximately"

61. P8L21. "boundary points", Modified from "boundary"

62. P8L23. "from which", Modified from "by which"

63. P9L3. "which linearly interpolated the high-resolution surface", Modified from "which linearly interpolated the high-resolution surface literally."

64. P9L6. "will result in", Modified from "warrants"

65. P9L14. "the", Added

66. P9L14. "dataset, Added

67. P9L18. "labelled", Modified from "labelled as"

68. P9L18. "so", Added

69. P9L20. "between", Added

70. P9L20. "southeastern", Modified from "south eastern"

71. P9L22. "northwestern", Modified from "north eastern"

72. P9L23. "due to", Modified from "for"

73. P9L25. "variations", Modified from "variation"

74. P9L27. "of", Added

75. P10L2. "As previously mentioned", Modified from "As aforementioned"

76. P10L2. "and", Modified from "with"

77. P10L2. "have been", Added

78. P10L8. "according to", Modified from "by"

79. P10L9. "highest", Modified from "largest"

80. P10L9. "modified", Modified from "modified by"

81. P10L10. "repeats", Modified from "loops"

82.P10L21. "Error estimates for", Modified from "Error estimates of"

83. P10L26. "the", Added

84. P10L27. "obtained", Modified from "from"

85. P10L27. "In all cases", Modified from "While all"

86. P10L30. "from the aspect of terrain structure retention.", Modified from "from terrain structure retention aspects"

87. P10L30. "resulting", Modified from "result"

88. P10L31. "datasets", Modified from "datesets"

89. P10L31. "performed well based on visual examination", Modified from "performed well from visual examination"

90. P11L8. "detail losses by", Modified from "detail loss from"

91. P11L10. "and", Added

92. P11L11. "For", Modified from "Upon"

93. P11L13. "overlaid", Modified from "overlapped"

94. P11L14. "cCVT-generalised surface accurately conform with", Modified from "cCVT-generalised surface are more accurately conform with"

95. P11L17. "near", Modified from "to"

96. P11L20. "Topography transformation of DEM surfaces has been a deeply studied topic in geoscience, simplification techniques and generalisation principles are widely realised and adopted. Extracting terrain feature points and using these points to construct a generalised surface has proven to be one valuable approach;"
Correspond to Change #25

97. P11L24. "Take the mountain equation in Figure 1 as an example", Modified from "Taking the mountain equation in Figure 1 for example"

98. P11L25. "the", Added

99. P11L25. "Assume that the", Modified from "Presume"

100. P11L27. "This observation implies that if global surface interpolation precision is of more importantly demanded, a robust approach that has overall considerations on surface approximation and terrain feature retention should be adopted"
Correspond to Change #26

101. P12L1. "However, the impact of the insertion of a new feature point on the inserted feature points is not considered due to computational burden. That is, modifications are only taken place on the triangle where the point with largest vertical error locates on. As a result, feature points may cluster around reliefs with sharp variations"
Correspond to Change #27

102. P12L10. "cCVT starts by constructing a terrain-adaptive variable-resolution grid. The cCVT iteration uses a robust mean curvature as density function which is based on the curvature's capability to characterise shapes and conduct shape evolution. Under the curvature guidance, the two-step optimisation (c.f. Algorithm1 in Section 2.2) loops both to spatial equalisation and frequency equalisation. The process of spatial equalising of feature points has been seldom considered by classical approaches, which may explain why cCVT generally prevails over HFPRs (c.f. Table 1). Notably, the triangles from the spatial equalisation exhibited a maximum aspect ratio that was less than 5.0, which implies that the constructed terrain grid satisfied the numerical stability requirement from classical finite element or finite volume computations."
Correspond to Change #28

103. P12L17. "On the other hand, CVT is an approach within variational framework. The result of the iteration depends on the boundary conditions and initial conditions. Hence, this article employed a feature point scheme (with additional input points considered) as a relatively stationary initial condition to maintain algorithm stability. The requirement of embedding feature points of interest, along with consideration of avoiding the problematic k-means like clustering, prompted us to develop a non-clustering approach with an exact energy referring method. Experiments on ten million DEM points demonstrated that the exact clipping approach performed comparably to the elegant clustering approach".
Correspond to Change #29

104. P12L24. "In this article, a high-fidelity multiresolution DEM model was proposed. The variable-resolution with high-fidelity was achieved by the developed curvature-based CVT. cCVT optimisation increases the precision of surface approximation compared to existing heuristic DEM generalisations, while the equalisation of feature points from the spatial domain guarantees a high-quality grid".
Correspond to Change #30

105. P12L28. "Multiresolution models are essential tools to incorporate more scales, while a high-fidelity generalised DEM model can be used to construct a concrete topographic layer from which fine endogenous or exogenous processes can be assessed under proper scale conditions. Evaluation of the cCVT multiresolution DEM model on Earth and environmental systems in wide-ranged domains and scales is a topic for future studies. Furthermore, considering the tyranny of global modelling of the Earth systems (Ringler et al., 2011), this may imply a consideration of curvature of the Earth itself into the cCVT model".
Correspond to Change #31

106. P13L2. "We implement the cCVT and the comparing classical k-means clustering CVT algorithm using c++ on the open-source Visualisation Toolkit (The VTK, https://www.vtk.org). The approximated dual operation through edge bisectioning is based on the implementation of Valette et al. (2008). If anyone is interested in the technical details of the implementation, please contact the corresponding author for the source codes, demo datasets and necessary guide for compilation".
Response to Special Comment about Code Availability

107. P18, Caption for Figure 1. "depiction", Modified from "depicting"

108. P19, Caption for Figure 3. "The triangles incident towards the first vertex", Modified from "The incident triangles toward"

109. P20, Caption for Figure 7. "and the projection on oriPd of the newly computed site", Modified from "the newly computed site"

110. P22, Caption for Figure 8. "Comparison of converged results", Modified from "converged results comparison"

111. P26, Caption for Figure 14, "close-up view", Modified from "close view"

112. P27, Caption for Figure 15, "was", Modified from "were"

[Revised Manuscript with Mark-up]

[revised manuscript text omitted]

批注 [o11]: Response to Comment #5

批注 [o12]: Correspond to Change #23

批注 [o13]: Response to Comment #6

批注 [s14]: Response to Comment #7

批注 [s15]: Response to Comment #8

批注 [s16]: Response to Comment 9#

批注 [s17]: Modified from "for"

[revised manuscript text omitted]

---

## Author Response (AR3)

**Response to Editor comments**

We would like to appreciate Topical Editor Dr. Jeffrey Neal for his careful work to the manuscript. We have carefully read the comments and made modifications accordingly. The responds to the comments and the modifications to the paper are as following:
(Review comments are reported in red.)

Comment#1.
Change
"Although there are many situations where dynamic conditions are stressed for stronger impacts on modifying predictions"
to
"Although there are many situations where dynamic conditions are stressed as stronger impacts on predictions"

**Response:**
We have modified it as the suggestion:
"Although there are many situations where dynamic conditions are stressed as stronger impacts on predictions"

Comment#2.
Change
"Evaluation of the cCVT multiresolution DEM model on Earth and environmental systems in wide ranged domains and scales is a topic for future studies."
to
Evaluation of the cCVT multiresolution DEM model on Earth and environmental systems over a wide range of domains and scales is a topic for future studies.

**Response:**
We have modified it as the suggestion:
"Evaluation of the cCVT multiresolution DEM model on Earth and environmental systems over a wide range of domains and scales is a topic for future studies."

Comment#3.
Finally, you need to give some context to the final sentence of the conclusions. I believe Ringler et al. are describing the tyranny of chasing ever finer resolution in earth system models at great computational cost. But this would not be clear from the sentence because tyranny could mean almost anything (for example that Earth system modelling is a bad thing per say), so this sentence would not convey the good point you wish to make.

**Response:**
Thanks to the Editor's remind, we here initially want to say when broader scales are involved, cCVT might be modified where geographical coordinate systems, geometric base (for clipping algorithm) and the curvature of the Earth itself

should be considered. (Now we realized that the curvature of the Earth itself might not be of so concerned). We modified this sentence to follow its initial intention as:

[revised manuscript text omitted]